# Sparse-Input Neural Network using Group Concave Regularization

**Bin Luo**                                                                                    *bluo@kennesaw.edu*
*School of Data Science and Analytics*
*Kennesaw State University*
*Marietta, GA 30060, USA*

**Susan Halabi** *                                                                     *susan.halabi@duke.edu*
*Department of Biostatistics and Bioinformatics*
*Duke University*
*Durham, NC 27708, USA*

**Reviewed on OpenReview:** *https://openreview.net/forum?id=m9UsLHZYeX*

## Abstract

Simultaneous feature selection and non-linear function estimation is challenging in modeling, especially in high-dimensional settings where the number of variables exceeds the available sample size. In this article, we investigate the problem of feature selection in neural networks. Although the group least absolute shrinkage and selection operator (LASSO) has been utilized to select variables for learning with neural networks, it tends to select unimportant variables into the model to compensate for its over-shrinkage. To overcome this limitation, we propose a framework of sparse-input neural networks using group concave regularization for feature selection in both low-dimensional and high-dimensional settings. The main idea is to apply a proper concave penalty to the $l_2$ norm of weights from all outgoing connections of each input node, and thus obtain a neural net that only uses a small subset of the original variables. In addition, we develop an effective algorithm based on backward path-wise optimization to yield stable solution paths, in order to tackle the challenge of complex optimization landscapes. We provide a rigorous theoretical analysis of the proposed framework, establishing finite-sample guarantees for both variable selection consistency and prediction accuracy. These results are supported by extensive simulation studies and real data applications, which demonstrate the finite-sample performance of the estimator in feature selection and prediction across continuous, binary, and time-to-event outcomes.

## 1 Introduction

Over the past few decades, advancements in molecular, imaging, and other laboratory tests have led to a growing interest in high-dimensional data analysis (HDDA) (Donoho et al., 2000). This type of data involves a large number of observed variables relative to the small sample size, which presents a considerable challenge in building accurate and interpretable models. For example, in bioinformatics, hundreds of thousands of RNA expressions, genome-wide association study (GWAS) data, and genomic data are used to understand disease biology and the correlation with clinical outcomes, with only hundreds of patients involved (Visscher et al., 2012; Hertz et al., 2016; Kim & Halabi, 2016; Beltran et al., 2017; Sumiyoshi et al., 2024). To address the curse of dimensionality, feature selection is a critical step in HDDA modeling. By identifying the most relevant features that capture the relationship with clinical outcomes, feature selection enhances model interpretability and improves generalization.

---

*Corresponding author.

There are various methods for feature selection, including filter methods (Koller & Sahami, 1996; Guyon & Elisseeff, 2003; Gu et al., 2012), wrapper methods (Kohavi & John, 1997; Inza et al., 2004; Tang et al., 2014), and embedded methods (Tibshirani, 1996; Zou, 2006; Fan & Li, 2001; Zhang et al., 2010). Among them, penalized regression methods have become very popular in HDDA since the introduction of the least absolute shrinkage and selection operator (LASSO) (Tibshirani, 1996). Penalized regression methods can perform simultaneous parameter estimation and feature selection by shrinking some of the parameter coefficients to exact zeros. While LASSO has been widely used to obtain sparse estimates in machine learning and statistics, it tends to select unimportant variables to compensate for the over-shrinkage for relevant variables (Zou, 2006). To address the bias and inconsistent feature selection of LASSO, several methods have been proposed, including adaptive LASSO (Zou, 2006), the minimax concave penalty (MCP) (Zhang et al., 2010), and the smoothly clipped absolute deviation (SCAD) (Fan & Li, 2001).

However, most of these penalized methods assume linearity in the relationship between the variables and the outcomes, while the actual functional form of the relationship may not be available in many applications. Some additive non-parametric extensions have been proposed to resolve this problem (Lin & Zhang, 2006; Ravikumar et al., 2009; Meier et al., 2009), but their models rely on sums of univariate or low-dimensional functions and may not be able to capture the complex interactions between multiple covariates. Yamada et al. (2014) propose the HSIC-LASSO approach that leverages kernel learning for feature selection while uncovering non-linear feature interactions. However, it suffers from quadratic scaling in computational complexity with respect to the number of observations.

Neural networks are powerful tools for modeling complex relationships in a wide range of applications, from imaging (Krizhevsky et al., 2017; He et al., 2016) and speech recognition (Graves et al., 2013; Chan et al., 2016) to natural language processing (Young et al., 2018; Devlin et al., 2018) and financial forecasting (Fischer & Krauss, 2018). Their state-of-the-art performance has been achieved through powerful computational resources and the use of large sample sizes. Despite that, high-dimensional data can still lead to overfitting and poor generalization performance for neural networks (Liu et al., 2017).

Recently, novel approaches have been developed that use regularized neural networks for feature selection or HDDA. A line of research focuses on utilizing the regularized neural networks, specifically employing the group LASSO technique to promote sparsity among input nodes (Liu et al., 2017; Scardapane et al., 2017; Feng & Simon, 2017). These methods treat all outgoing connections from a single input neuron as a group and apply the LASSO penalty to the $l_2$ norm of weight vectors of each group. Other LASSO-regularized neural networks in feature selection can be found in the work of Li et al. (2016) and Lemhadri et al. (2021). However, LASSO-regularized neural networks tend to over-shrink the weights of relevant variables, leading to the inclusion of many false positives. The adaptive LASSO was employed to alleviate this problem (Dinh & Ho, 2020), yet their results are limited to continuous outcomes and assume that the conditional mean function is exactly a neural network. The work in Yamada et al. (2020) bypassed the $l_1$ regularization by introducing stochastic gates to the input layer of neural networks. They considered $l_0$-like regularization based on a continuous relaxation of the Bernoulli distribution. Their method, however, requires a cutoff value for selecting variables with weak signals, and the stochastic gate is unable to completely exclude the non-selected variables during model training and prediction stages.

Recent theoretical advancements have provided rigorous guarantees for sparse deep learning. For instance, Sun et al. (2022) proposed a method justified under a Bayesian framework that can learn a sparse deep neural network with theoretical guarantees for posterior consistency and variable selection consistency. Their approach first trains a dense network with a mixture Gaussian prior and then sparsifies its structure using a Laplace approximation of marginal posterior inclusion probabilities. Building on a different theoretical perspective, Lederer (2024) developed statistical guarantees for sparse deep learning from a high-dimensional statistics perspective, providing oracle inequalities that guarantee near-optimal prediction accuracy for various types of network sparsity, including connection and node sparsity. Despite this progress, however, theory that provides direct frequentist guarantees for variable selection consistency of penalized estimators, a cornerstone of classical high-dimensional statistics, remains an important area for development in deep learning.

In this paper, we propose a novel framework for sparse-input neural networks using group concave regularization to overcome the limitations of existing feature selection methods. Although folded concave penalties like MCP and SCAD have been shown to perform well in both theoretical and numerical settings for feature selection and prediction, they have not received the same level of attention as LASSO in the context of machine learning. Our proposed framework aims to draw attention to the underutilized potential of concave penalties for feature selection in neural networks by providing a comprehensive approach for simultaneous feature selection and function estimation in both low- and high-dimensional settings.

The key contributions of this paper are as follows:

- A unified framework for simultaneous feature selection and prediction: We introduce structured sparsity in neural networks by applying concave group penalties, treating all outgoing connections from a single input neuron as a group. An $l_2$-norm-based concave penalty shrinks entire groups of weights to zero, resulting in a parsimonious neural network that selects only a small subset of input variables.

- An effective optimization strategy for stable solution paths: We employ backward path-wise optimization, a dense-to-sparse approach that traces the solutions of the regularized neural network. This approach improves the stability of model selection and enhances the interpretability of the solution path across the full range of regularization parameters.

- We provide a rigorous theoretical analysis of our estimator, establishing non-asymptotic bounds on its prediction and estimation error and proving that it possesses the desirable oracle property under standard high-dimensional conditions.

- Empirical validation across diverse data types: Through extensive simulations and real-data analysis, we demonstrate that our method outperforms existing approaches in feature selection consistency and prediction accuracy across continuous, binary, and time-to-event outcomes.

The rest of this article is organized as follows. In Section 2, we formulate the problem of feature selection for a generic non-parametric model and introduce our proposed method. The implementation of the method, including the composite gradient descent algorithm and the backward path-wise optimization, is presented in Section 3. In Section 4, we establish the theoretical properties of our proposed estimator. In Section 5, we conduct extensive simulation studies to demonstrate the performance of the proposed method. In Section 6, we apply the proposed method to real-world datasets. Lastly, in Section 7, we discuss the results and their implications. The theoretical proofs, implementation details, and supplementary numerical results are provided in the Appendix.

## 2  Method

### 2.1  Problem setup

Let $X \in \mathbb{R}^d$ be a $d$-dimensional random vector and $Y$ be a response variable. We assume the conditional distribution $P_{Y|X}$ depends on a form of $f(X_S)$ with a function $f \in F$ and a subset of variables $S \subseteq \{1, \cdots, d\}$. We are interested in identifying the true set $S$ for important variables and estimating function $f$ so that we can predict $Y$ based on selected variable $X_S$.

At the population level, we aim to minimize the loss

$$\min_{f \in F, S} \mathbb{E}_{X,Y} \ell(f(X_S), Y),$$

where $\ell$ is a loss function tailored to a specific problem. In practical settings, the distribution of $(X, Y)$ is often unknown, and instead only an independent and identically distributed (i.i.d.) random sample of size $n$ is available, consisting of pairs of observations $(X_i, Y_i)_{i=1}^n$. Additionally, if the number of variables $d$ is large, an exhaustive search over all possible subsets $S$ becomes computationally infeasible. Furthermore, we do not assume any specific form of the unknown function $f$ and aim to approximate $f$ nonparametrically

using neural networks. Thus, our goal is to develop an efficient method that can simultaneously select a variable subset $S$ and approximate the solution $f$ for any given class of functions using a sparse-input neural network.

## 2.2 Proposed framework

We consider function estimators based on feedforward neural networks. Let $\mathcal{F}_n$ be a class of feed forward neural networks $f_{\mathbf{w}} : \mathbb{R}^d \mapsto \mathbb{R}$ with parameter $\mathbf{w}$. The architecture of a multi-layer perceptron (MLP) can be expressed as a composition of a series of functions

$$f_{\mathbf{w}}(x) = L_D \circ \sigma \circ L_{D-1} \circ \sigma \circ \cdots \circ \sigma \circ L_1 \circ \sigma \circ L_0(x), x \in \mathbb{R}^d,$$

where $\circ$ denotes function composition and $\sigma(x)$ is an activation function defined for each component of $x$. Additionally,

$$L_i(x) = \mathbf{W}_i x + b_i, i = 0, 1, \ldots, \mathcal{D},$$

where $\mathbf{W}_i \in \mathbb{R}^{d_{i+1} \times d_i}$ is a weight matrix, $\mathcal{D}$ is the number of hidden layers, $d_i$ is the width defined as the number of neurons of the $i$-th layer with $d_0 = d$, and $b_i \in \mathbb{R}^{d_{i+1}}$ is the bias vector in the $i$-th linear transformation $L_i$. Note that the vector $\mathbf{w} \in \mathbb{R}^p$ is the column-vector concatenation of all parameters in $\{\mathbf{W}_i, b_i : i = 0, 1, \ldots, \mathcal{D}\}$. We define the empirical loss of $f_{\mathbf{w}}$ as

$$\mathcal{L}_n(\mathbf{w}) = \frac{1}{n} \sum_{i=1}^{n} \ell(f_{\mathbf{w}}(X_i), Y_i).$$

Ideally, the sparse-input neural network $f_{\mathbf{w}}$ should rely only on the important variables, meaning that $\mathbf{W}_{0,j} = \mathbf{0}$ for $j \notin S$, where $\mathbf{W}_{0,j}$ denotes the $j$th column vector of $\mathbf{W}_0$. In order to minimize the empirical loss $\mathcal{L}_n(\mathbf{w})$ while inducing sparsity in $\mathbf{W}_0$, we propose to train the neural network by minimizing the following group regularized empirical loss

$$\hat{\mathbf{w}} = \arg\min_{\mathbf{w} \in \mathbb{R}^p} \left\{ \mathcal{L}_n(\mathbf{w}) + \sum_{j=1}^{d} \rho_\lambda(\|\mathbf{W}_{0,j}\|_2) + \alpha\|\mathbf{w}\|_2^2 \right\}, \tag{1}$$

where $\|\cdot\|_2$ denotes the Euclidean norm of a vector.

The objective function in Eq. (1) comprises three components:

(1) $\mathcal{L}_n(\mathbf{w})$ is the empirical loss function, such as the mean squared error loss for regression tasks, the cross-entropy loss for classification tasks, and the negative log partial likelihood for proportional hazards models. Further details can be found in Appendix A.

(2) $\rho_\lambda$ is a concave penalty function parameterized by $\lambda \geq 0$. To simultaneously select variables and learn the neural network, we group the outgoing connections from each single input neuron that corresponds to each variable. The concave penalty function $\rho_\lambda$ is designed to shrink the weight vectors of specific groups to exact zeros, resulting in a neural network that utilizes only a small subset of the original variables.

(3) The term $\alpha\|\mathbf{w}\|_2^2$ is a ridge regularization that plays a critical role in ensuring both model stability and the effectiveness of the feature selection mechanism. Note that the group penalty $\rho_\lambda$ acts only on the input layer weights $\mathbf{W}_0$. Without the ridge penalty ($\alpha = 0$), a neural network could circumvent this penalty by shrinking $\mathbf{W}_0$ while inflating weights in subsequent layers to compensate, leaving the network output unchanged and rendering feature selection ineffective. The ridge penalty prevents this compensatory behavior by regularizing the norms of all weights in the network ($\mathbf{w}$), encouraging smaller and well-balanced weights throughout the architecture. Moreover, in Section 4, the ridge term is expressed as a norm constraint on the full parameter vector, which is essential for deriving our statistical guarantees.

It should be noted that when the number of hidden layers $D = 0$, the function $f_{\mathbf{w}}$ reduces to a linear function, and the optimization problem in Eq. (1) becomes the framework of elastic net (Zou & Hastie, 2005), SCAD-$L_2$ (Zeng & Xie, 2014), and Mnet (Huang et al., 2016), with the choice of $\rho_\lambda$ to be LASSO, SCAD, and MCP, respectively.

## 2.3 Concave regularization

There are several commonly used penalty functions that encourage sparsity in the solution, such as LASSO (Tibshirani, 1996), SCAD (Fan & Li, 2001), and MCP (Zhang et al., 2010). When applied to the $l_2$-norm of the coefficients associated with each group of variables, these penalty functions give rise to group regularization methods, including group LASSO (GLASSO) (Yuan & Lin, 2006), group SCAD (GSCAD) (Guo et al., 2015), and group MCP (GMCP) (Huang et al., 2012). Specifically, LASSO, SCAD, and MCP are defined as follows.

- **LASSO**

$$\rho_\lambda(t) = \lambda |t|.$$

- **SCAD**

$$\rho_\lambda(t) = \begin{cases} \lambda|t| & \text{for } |t| \leq \lambda, \\ -\frac{t^2 - 2a\lambda|t| + \lambda^2}{2(a-1)} & \text{for } \lambda < |t| \leq a\lambda, \\ \frac{(a+1)\lambda^2}{2} & \text{for } |t| > a\lambda, \end{cases}$$

  where $a > 2$ is fixed.

- **MCP**

$$\rho_\lambda(t) = \text{sign}(t)\lambda \int_0^{|t|} \left(1 - \frac{z}{\lambda a}\right)_+ dz,$$

  where $a > 0$ is fixed.

It has been demonstrated, both theoretically and numerically, that the folded concave regularization methods of SCAD and MCP exhibit strong performance in terms of feature selection and prediction (Fan & Li, 2001; Zhang et al., 2010). Unlike the convex penalty LASSO, which tends to over-regularize large terms and provide inconsistent feature selection, concave regularization can reduce LASSO's bias and improve model selection accuracy. The rationale behind the concave penalty lies in the behavior of its derivatives. Specifically, SCAD and MCP initially apply the same level of penalization as LASSO, but gradually reduce the penalization rate until it drops to zero when $t > a\lambda$. Given the benefits of the concave penalization, we propose using the group concave regularization in our framework for simultaneous feature selection and function estimation.

## 3 Implementation

### 3.1 Composite gradient descent

The optimization in Eq. (1) is not a convex optimization problem since both empirical loss function $\mathcal{L}_n(\mathbf{w})$ and the penalty function $\rho_\lambda$ can be non-convex. To obtain the stationary point, we use the composite gradient descent algorithm (Nesterov, 2013). This algorithm is also incorporated in Feng & Simon (2017); Lemhadri et al. (2021) for sparse-input neural networks based on the LASSO regularization. The local convergence of the composite gradient descent algorithm for nonconvex regularization was established in Gong et al. (2013).

Denote $\bar{\mathcal{L}}_{n,\alpha}(\mathbf{w}) = \mathcal{L}_n(\mathbf{w}) + \alpha\|\mathbf{w}\|_2^2$ as the smooth component of the objective function in Eq. (1). The composition gradient iteration for epoch $t$ is given by

$$\mathbf{w}^{t+1} = \arg\min_{\mathbf{w}} \left\{ \frac{1}{2}\|\mathbf{w} - \tilde{\mathbf{w}}^{t+1}\|_2^2 + \gamma \sum_{j=1}^{d} \rho_\lambda(\|\mathbf{W}_{0,j}\|_2) \right\}, \tag{2}$$

where $\tilde{\mathbf{w}}^{t+1} = \mathbf{w}^t - \gamma\nabla\bar{\mathcal{L}}_{n,\alpha}(\mathbf{w}^t)$ is the gradient update only for the smooth component $\bar{\mathcal{L}}_{n,\alpha}(\mathbf{w}^t)$ that can be computed using the standard back-propagation algorithm. Here $\gamma > 0$ is the step size for the update and can be set as a fixed value or determined by employing the backtracking line search method, as described in Nesterov (2013). Let $A_j$ represent the index set of $\mathbf{W}_{0,j}$ within $\mathbf{w}$. We define $A$ as the index set that includes all weights in the input layer, given by $A = \{\bigcup_{j=1}^d A_j\}$. By solving Eq. (2), we obtain the iteration form $\mathbf{w}_{A^c}^{t+1} = \tilde{\mathbf{w}}_{A^c}^{t+1}$ and

$$\mathbf{w}_{A_j}^{t+1} = h(\tilde{\mathbf{w}}_{A_j}^{t+1}; \gamma, \lambda), \quad \text{for } j = 1, \cdots, d. \tag{3}$$

Here, $A^c$ refers to the complement of the set $A$, and the function $h$ represents the thresholding operator, which can be determined by the specific penalty $\rho_\lambda$. By taking $\rho_\lambda$ to be the LASSO, MCP, and SCAD penalty, it can be verified that the GLASSO, GSCAD, and GMCP solutions for the iteration in Eq. (3) have the following form:

- GLASSO

$$h_{\text{GLASSO}}(z; \gamma, \lambda) = S_g(z, \gamma\lambda).$$

- GSCAD

$$h_{\text{GSCAD}}(z; \gamma, \lambda) = \begin{cases} S_g(z, \gamma\lambda), & \text{if } \|z\|_2 \leq (\gamma+1)\lambda, \\ \frac{a-1}{a-1-\gamma}S_g(z, \frac{a\gamma\lambda}{a-1}), & \text{if } (\gamma+1)\lambda < \|z\|_2 \leq a\lambda, \\ z, & \text{if } \|z\|_2 > a\lambda. \end{cases}$$

- GMCP

$$h_{\text{GMCP}}(z; \gamma, \lambda) = \begin{cases} \frac{a}{a-\gamma}S_g(z, \gamma\lambda), & \text{if } \|z\|_2 \leq a\lambda, \\ z, & \text{if } \|z\|_2 > a\lambda, \end{cases}$$

where $S_g(z; \lambda)$ is the group soft-thresholding operator defined as

$$S_g(z; \lambda) = \left(1 - \frac{\lambda}{\|z\|_2}\right)_+ z.$$

Therefore, we can efficiently implement the composite gradient descent by integrating an additional thresholding operation into the input layer. This operation follows the gradient descent step using the smooth component $\bar{\mathcal{L}}_{n,\alpha}(\mathbf{w})$. The calculation for epoch t can be summarized as follows:

(1) Compute the gradient of the smooth component $\nabla\bar{\mathcal{L}}_{n,\alpha}(\mathbf{w}^t)$ using back-propagation.

(2) Perform the gradient update for the smooth component to get an intermediate estimate:

$$\tilde{\mathbf{w}}^{t+1} \leftarrow \mathbf{w}^t - \gamma\nabla\bar{\mathcal{L}}_{n,\alpha}(\mathbf{w}^t).$$

(3) Apply the thresholding operator to obtain the updated weights $\mathbf{w}^{t+1}$:

$$\mathbf{w}_{A^c}^{t+1} \leftarrow \tilde{\mathbf{w}}_{A^c}^{t+1}, \quad \mathbf{w}_{A_j}^{t+1} \leftarrow h\left(\tilde{\mathbf{w}}_{A_j}^{t+1}; \gamma, \lambda\right), \quad \text{for } j = 1, \ldots, d.$$

The final index set of the selected variables is $\hat{S} = \{j : \hat{\mathbf{w}}_{A_j} \neq \mathbf{0}\}$. Note that we consider $\gamma$ as a scaling factor in the thresholding operator. When $\gamma = 1$ in Step (3), the solutions for GLASSO, GSCAD, and GMCP align with the closed-form results established in Wei & Zhu (2012).

## 3.2 Backward path-wise optimization

We are interested in learning neural networks not only for a specific value of $\lambda$, but also for a range of $\lambda$s where the networks vary by the number of included variables. Specifically, we consider a range of $\lambda$ from $\lambda_{min}$, where the networks include all or an excessively large number of variables, up to $\lambda_{max}$, where all variables are excluded and $|W_0|$ becomes a zero matrix. Since the objective function is not convex and has

multiple local minima, the solution of Eq. (1) with random initialization may not vary continuously for $\lambda \in [\lambda_{min}, \lambda_{max}]$, resulting in a highly unstable path of solutions that are regularized by $\lambda$.

To address this issue, we consider a path-wise optimization strategy by varying the regularization parameter along a path. The goal is to generate a stable solution path, which we define as a sequence of solutions where small changes in the regularization parameter $\lambda$ result in correspondingly small changes in the model's weights and sparsity level, avoiding the large, erratic jumps often seen with random initializations for each $\lambda$ (see Figure 1). It is important to clarify that the stability of the path is a product of this overall strategy, not a property of the optimizer (e.g., Adam) used for any single value of $\lambda$. In this approach, we use the solution of a particular value of $\lambda$ as a warm start for the next problem. Regularized linear regression methods (Friedman et al., 2007; 2010; Breheny & Huang, 2011) typically adopt a forward path-wise optimization, starting from a null model with all variables excluded at $\lambda_{max}$ and working forward with decreasing $\lambda$s. However, our numerical studies on sparse-input neural networks revealed that starting with a sparse solution as the initial model does not lead to a smooth transition to a dense model. To address this challenge, we implement a backward path-wise optimization approach, starting from a dense model at the minimum value of $\lambda_{min}$ and solving toward sparse models up to $\lambda_{max}$ with all variables excluded from the network. This dense-to-sparse warm start approach is also employed in (Lemhadri et al., 2021) using LASSO regularization.

To further illustrate the importance of using backward path-wise optimization in regularized neural networks, we investigate variables selection and function estimation of a regression model $Y = f(X) + \epsilon$, where $f(X) = \log(|X_1| + 0.1) + X_1 X_2 + X_2 + \exp(X_3 + X_4)$ with 4 informative and 16 nuisance variables, and each $X_i$ and $\epsilon$ follow the standard normal distribution. We present more details of the simulations in Section 5. Figure 1 shows the solution paths of GSCAD, GMCP, and GLASSO based on different types of optimization. In these plots, each line represents the path of the group weight norm for a single input variable; green lines indicate the four true informative variables and gray dashed lines indicate the sixteen nuisance variables. It is observed that non-pathwise optimization (top-left) leads to fluctuations or variations in the solution path, whereas forward path-wise optimization tends to remain in the same sparse model (GMCP, top-middle) or experience fluctuation solutions (GLASSO, top-right), until transitioning to a full model with a sufficiently small $\lambda$. In contrast, backward path-wise optimization (bottom panels) yields smoother solution paths for informative and nuisance variables. Importantly, GLASSO (bottom-right) tends to over-shrink the weight vectors of informative variables and include more variables in the model, while GSCAD (bottom-left) and GMCP (bottom-middle) are designed to prevent over-shrinkage and offer a smooth transition from the full to the null model.

In addition to providing stable and smooth solution paths, backward path-wise optimization is advantageous computationally. In particular:

- The consecutive estimates of weights in the path are close, which reduces the number of gradient descent iterations required for each $\lambda$. Therefore, the bulk of the computational cost occurs at $\lambda_{min}$, and a lower number of iterations for the remaining $\lambda$s results in low computational costs.

- We observe that the excluded variables from previous solutions are rarely included in the following solutions. By pruning the inputs of the neural network along the solution path, further reduction in computation complexity can be achieved as the model becomes sparse. Since the computational cost scales with the number of input features, this approach can significantly speed up computation, particularly for high-dimensional data.

### 3.3 Tuning Parameter Selection

Two tuning parameters are required in our proposed framework: the group penalty coefficient $\lambda$ and the ridge penalty coefficient $\alpha$. The former controls the number of selected variables and yields sparser models for larger values of $\lambda$, while the latter imposes a penalty on the size of the network weights to prevent overfitting.

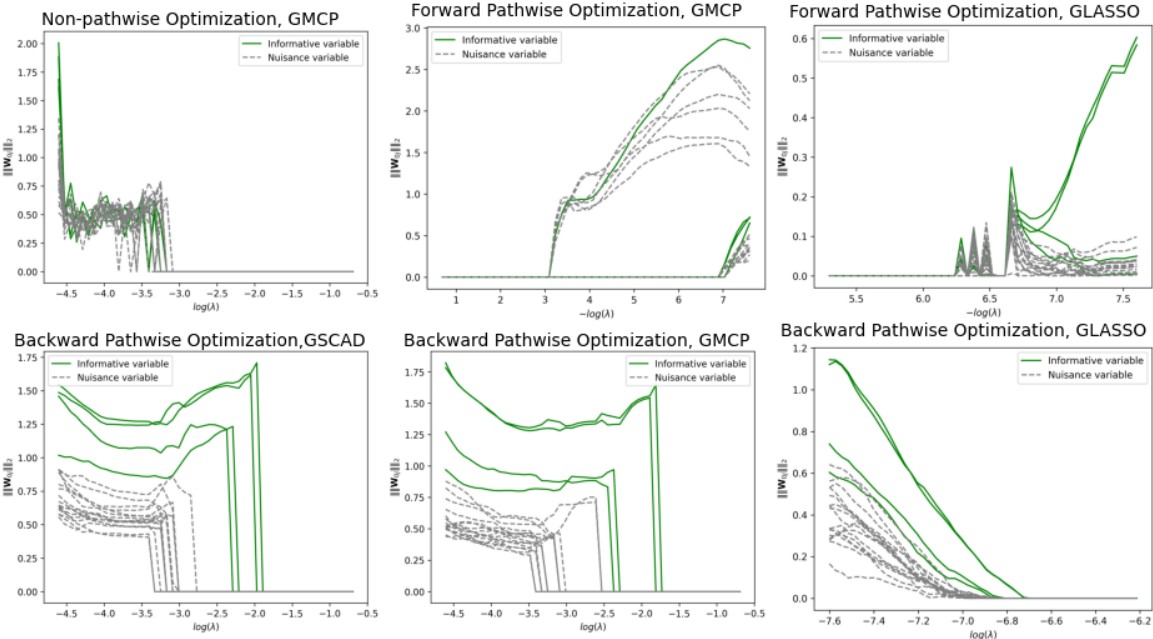

Figure 1: **Solution path of $l_2$ norm of the weight vector associated with each input node $\|\mathbf{W}_{0j}\|_2$.** Each line shows the path for a single variable's group weight norm; green lines correspond to the four true informative variables and gray dashed lines to the sixteen nuisance variables. The bottom panels illustrate the "smoother" solution paths generated by our backward pathwise strategy, where weight norms change predictably without the erratic jumps seen in the non-pathwise (top left) and forward pathwise (top middle, right) approaches. Note that the x-axis, $\log(\lambda)$, differs between plots because the absolute scale of the regularization parameter is not directly comparable across penalty types. The individual panel descriptions are as follows: **Top left**: Non-pathwise optimization using GMCP. All the neural network weights are initialized by drawing from $N(0, 0.1)$ for each $\lambda$. **Top middle**: forward path-wise optimization using GMCP. It starts from the null model and computes the solution with decreasing $\lambda$. Random initialization is used before the selection of the first set of variables. **Top right**:forward path-wise optimization using GLASSO. **Bottom left**: backward path-wise optimization using GSCAD. **Bottom middle**: backward path-wise optimization using GMCP. **Bottom right**: backward path-wise optimization using GLASSO.

In all numerical studies presented in this paper, we adopted a 20% holdout validation set from the training data. The model was trained using the remaining data, and the optimal values for $\lambda$ and $\alpha$ were selected from a fine grid of values based on their performance on the validation set.

Python code and examples for the proposed group concave regularized neural networks are available at https://github.com/r08in/GCRNN.

# 4 Theoretical Properties

In this section, we establish the theoretical underpinnings of our proposed estimator, providing non-asymptotic, finite-sample guarantees on its performance. Our primary goal is to show that, under standard high-dimensional regularity conditions, our method achieves strong statistical properties in terms of both prediction accuracy and variable selection consistency.

For the theoretical analysis, we constrain the norm of full vector $\mathbf{w}$ instead of using a ridge penalty. Our problem formulation for training the sparse-input neural network becomes:

$$\hat{\mathbf{w}} \in \arg\min_{\mathbf{w}\in\Theta} \left\{ \frac{1}{n}\sum_{i=1}^{n}\ell(f_{\mathbf{w}}(X_i), Y_i) + \sum_{j=1}^{d}\rho_\lambda(\|\mathbf{W}_{0,j}\|_2) \right\} \tag{4}$$

where $\Theta = \{\mathbf{w} \in \mathbb{R}^p : \|\mathbf{w}\|_2 \le K\}$ for a constant $K > 0$.

As is common in the literature for non-convex problems, our analysis focuses on the properties of a suitable local minimizer, leaving the computationally challenging issue of finding a global minimum to future work.

## 4.1 Preliminaries and Notation

Suppose covariates $X$ have support $\mathcal{X} \subseteq [-X_{max}, X_{max}]^d$ and $\mathcal{X}$ contains some open set. We assume the true data generating process depends on a function $f^*$ that uses only a subset of important variables $S \subseteq \{1, \ldots, d\}$. Let $\mathbb{E}$ denote the expectation with respect to the joint distribution of $(X, Y)$. Given $n$ i.i.d. observations, we denote the empirical expectation as $\mathbb{E}_n$.

Let the set of optimal neural networks that minimize the expected loss be denoted

$$EQ^* = \arg\min_{\mathbf{w}\in\Theta} \mathbb{E}[\ell(f_{\mathbf{w}}(X), Y)]. \tag{5}$$

Due to the symmetries inherent in neural network architectures (e.g., permutation of hidden nodes), the optimal parameter vector is not unique. Following the framework of Feng & Simon (2017), we assume the parameters in $EQ^*$ can be partitioned into a finite number of equivalence classes, denoted by $Q \ge 1$. For any $\mathbf{w}^* \in EQ^*$, we assume that all weight groups tied to the irrelevant features $S^c$ are zero (i.e., $\mathbf{W}^*_{0,j} = \mathbf{0}$ for all $j \in S^c$). For any parameter vector $\mathbf{w}$, let the closest element in $EQ^*$ be defined as $\mathbf{w}^{*(\mathbf{w})} \in \arg\min_{\mathbf{w}^* \in EQ^*} \|\mathbf{w} - \mathbf{w}^*\|_2$. We define the excess loss of a neural network with parameters $\mathbf{w}$ as

$$\mathcal{E}(\mathbf{w}) = \mathbb{E}[\ell(f_{\mathbf{w}}(X), Y)] - \min_{\mathbf{w}'\in\Theta} \mathbb{E}[\ell(f_{\mathbf{w}'}(X), Y)] \tag{6}$$

$$= \mathbb{E}[\ell(f_{\mathbf{w}}(X), Y) - \ell(f_{\mathbf{w}^{*(\mathbf{w})}}(X), Y)].$$

For our theoretical analysis, we partition the input layer's weight matrix, $\mathbf{W}_0$, column-wise according to the true support set $S$, which contains the indices of the $s = |S|$ important variables. This yields two sub-matrices: $\mathbf{W}_{0,S}$, with columns corresponding to variables in $S$, and $\mathbf{W}_{0,S^c}$, with columns for the irrelevant variables in $S^c$. We define $\mathbf{w}_{\text{upper}}$ as the vector of all parameters not subject to the group-concave penalty; this includes the input layer's bias vector ($b_0$) and all subsequent weights and biases, which has dimension $d_{\text{upper}}$. This partitioning defines the oracle vector $\mathbf{w}_S$, which combines all parameters of the true model structure ($\mathbf{W}_{0,S}$ and $\mathbf{w}_{\text{upper}}$) for a total dimension of $d^* = (s \times d_1) + d_{\text{upper}}$, as well as the full vector $\mathbf{w}$ with total dimension $p = (d \times d_1) + d_{upper}$. With a slight abuse of notation, we denote the oracle parameter support set of $\mathbf{w}_S$ by $S_{\text{param}}$, where $|S_{\text{param}}| = d^*$. This set is defined as the union of the index sets corresponding to $\mathbf{W}_{0,S}$ and $\mathbf{w}_{\text{upper}}$.

## 4.2 Statistical Guarantees for Estimation and Prediction

We now present our first main result, which provides deterministic upper bounds on the estimation and prediction performance of any local minimizer whose irrelevant weights lie within the active penalty region. To establish these guarantees, we rely on a set of standard technical conditions widely used in high-dimensional statistics. These assumptions ensure that the loss function is well-behaved around the true parameters, that the model is identifiable, and that our nonconvex penalty can induce sparsity without introducing excessive bias.

**Condition 4.1** (Local Strong Convexity). Let the parameter vector $\mathbf{w}$ be ordered such that the irrelevant weight groups $\mathbf{W}_{0,S^c}$ are first. There is a constant $h_{min} > 0$, which may depend on the network architecture, $s$, and $f^*$, but not on $d$, such that for all $\mathbf{w}^* \in EQ^*$,

$$\nabla^2_{\mathbf{w}} \mathbb{E}[\ell(f_{\mathbf{w}}(X), Y)]_{\mathbf{w}=\mathbf{w}^*} \geq h_{min} \begin{bmatrix} 0 & 0 \\ 0 & I \end{bmatrix}. \tag{7}$$

This condition guarantees that the loss function has sufficient curvature around the true solution with respect to the relevant parameters. This ensures a well-defined minimum, enabling stable and consistent estimation.

**Condition 4.2** (Compatibility Condition). For all $\epsilon > 0$, there is a $\chi_\epsilon > 0$ that does not depend on $d$, such that

$$\chi_\epsilon \leq \inf_{\mathbf{w} \in \Theta} \left\{ \mathcal{E}(\mathbf{w}) : ||\mathbf{w}_S - \mathbf{w}_S^{*(\mathbf{w})}||_2 \geq \epsilon \text{ and } \sum_{j \in S^c} ||\mathbf{W}_{0,j}||_2 \leq 5 \sum_{j \in S} ||\mathbf{W}_{0,j} - \mathbf{W}_{0,j}^{*(\mathbf{w})}||_2 + ||\mathbf{w}_{upper} - \mathbf{w}_{upper}^{*(\mathbf{w})}||_2 \right\},$$

where $\mathbf{w}_S$ and $\mathbf{w}_{\text{upper}}$ are as defined in Section 4.1.

This is a standard requirement for high-dimensional analysis that links parameter error to prediction error. It ensures that a meaningful deviation from the true parameters results in a quantifiable increase in the excess risk, which is crucial for model identifiability.

**Condition 4.3** (Bounded Third Derivative). The third derivative of the expected loss function is bounded uniformly over $\Theta$ by some constant $G > 0$ that does not depend on $d$:

$$\sup_{\mathbf{w} \in \Theta} \max_{j_1, j_2, j_3} \left| \frac{\partial^3}{\partial w_{j_1} \partial w_{j_2} \partial w_{j_3}} \mathbb{E}[\ell(f_{\mathbf{w}}(X), Y)] \right| \leq G. \tag{8}$$

This condition of bounded third derivative ensures the loss function is locally well-approximated by a quadratic. This regularity is required for the Taylor series arguments that underpin our proofs.

**Remark 4.1.** *Note that the bounded third-derivative condition is not strictly satisfied by non-smooth activations like ReLU (Nair & Hinton, 2010), which, although computationally efficient, is non-differentiable at the origin. Consequently, our theoretical guarantees apply most directly to the case of smooth activations, such as Softplus (Glorot et al., 2011) or GeLU (Hendrycks & Gimpel, 2016). In Section 5, we discuss the practical implementation using ReLU, where we observe empirical performance consistent with the theoretical properties established for smooth activations.*

**Condition 4.4** (Properties of the Nonconvex Penalty Function). The penalty is a scalar function $\rho_\lambda : [0, \infty) \mapsto [0, \infty)$ satisfies:

(i) $\rho_\lambda(0) = 0$.

(ii) The function $t \mapsto \rho_\lambda(t)$ is non-decreasing.

(iii) The function $t \mapsto \frac{\rho_\lambda(t)}{t}$ is non-increasing on $(0, \infty)$. This implies that $\rho_\lambda(t)$ is concave on $[0, \infty)$.

(iv) The function $t \mapsto \rho_\lambda(t)$ is differentiable on $(0, \infty)$.

(v) The derivative satisfies $\lim_{t \to 0^+} \rho'_\lambda(t) = \lambda$.

    (vi) There exists a constant $\delta > 0$ such that $\delta\lambda$ is the smallest value for which the derivative $\rho'_\lambda(t) = 0$ for all $t \geq \delta\lambda$.

    (vii) The derivative $t \mapsto \rho'_\lambda(t)$ is a concave function on the interval $[0, \delta\lambda]$.

These properties formalize the requirements for a penalty function that can produce sparse and nearly unbiased estimates. The key features are a sharp penalty at the origin to enforce sparsity and a vanishing penalty for large coefficients to reduce estimation bias. Common penalties that satisfy these properties include MCP and the SCAD.

Under these conditions, we are ready to state our main deterministic result, which provides explicit bounds on the estimation error, excess risk, and sparsity recovery for local minimizers of the penalized objective.

**Theorem 4.1.** *For any $\tilde{\lambda} > 0$ and $T \geq 1$ let*

$$\mathcal{T}_{\tilde{\lambda},T} = \{\{(x_i, y_i)\}_{i=1}^n : \sup_{\mathbf{w} \in \Theta} \frac{|(\mathbb{E}_n - \mathbb{E})(\ell(f_{\mathbf{w}^*(\mathbf{w})}(x), y) - \ell(f_{\mathbf{w}}(x), y))|}{\tilde{\lambda} \vee (||\mathbf{w}_{upper} - \mathbf{w}_{upper}^{*(\mathbf{w})}||_2 + \sum_{j=1}^d ||\mathbf{W}_{0,j} - \mathbf{W}_{0,j}^{*(\mathbf{w})}||_2)} \leq T\tilde{\lambda}\}.$$

*Suppose Conditions 4.1-4.4 hold. Let $\hat{\mathbf{w}}$ be a local minimizer of equation 4 such that $\max_{j \in S^c} ||\hat{\mathbf{W}}_{0,j}||_2 \leq \delta\lambda$. Then over the set $\mathcal{T}_{\tilde{\lambda},T}$, for any tuning parameter $\lambda \geq 4T\tilde{\lambda}$, such a local minimizer $\hat{\mathbf{w}}$ exists and satisfies*

    (i) **Estimation Error:** $||\hat{\mathbf{w}}_S - \mathbf{w}_S^*||_2 \leq C_0^2(\lambda + T\tilde{\lambda})\sqrt{1 + s}$.

    (ii) **Excess Risk:** $\mathcal{E}(\hat{\mathbf{w}}) \leq (\lambda + T\tilde{\lambda})^2(1 + s)C_0^2$.

    (iii) **Sparsity Recovery:** $\sum_{j \in S^c} ||\hat{\mathbf{W}}_{0,j}||_2 \leq \frac{(\lambda + T\tilde{\lambda})^2(1 + s)C_0^2}{\lambda - 2T\tilde{\lambda}}$.

*where*

$$C_0^2 = \frac{1}{\epsilon_0} \vee \frac{2K}{\chi_{\epsilon_0}} \tag{9}$$

$$\epsilon_0 = \frac{h_{min}}{\frac{2}{3}G(\sqrt{d_1}(1 + 5\sqrt{s}) + \sqrt{d^*})^3}. \tag{10}$$

The proof of Theorem 4.1 is included in Section B.1. Note that the statement of Theorem 4.1 is entirely deterministic and the established bounds are expressed in terms of the tuning parameter $\lambda$ and the statistical complexity term $\tilde{\lambda}$. By having $\lambda$ vanish at the same rate of $\tilde{\lambda}$, the estimation error, excess risk and the norm of irrelevant weights converges on the order of $O_p(\tilde{\lambda}s^2 d_1^{3/2}G)$, $O_p(\tilde{\lambda}^2 s^{5/2} d_1^{3/2}G)$, $O_p(\tilde{\lambda}s^{5/2} d_1^{3/2}G)$, respectively.

**Remark 4.2.** *Theorem 4.1 establishes statistical consistency only for local minimizers whose irrelevant weights lie within the active penalty region, i.e., $\max_{j \in S^c} ||\hat{\mathbf{W}}_{0,j}||_2 \leq \delta\lambda$. This requirement is not restrictive in practice: by suitably choosing $\delta$ or $\lambda$, one can either enlarge the active penalty region or drive the irrelevant weights toward zero, so that such a local minimizer may also coincide with a global minimizer. Moreover, it is preferable for the irrelevant weights to fall in this region, as this is the mechanism for inducing sparsity. In general, even if a global minimizer does not satisfy the condition, Theorem 4.1 guarantees the existence of a statistically consistent local minimizer that does.*

The bounds in Theorem 4.1 depend on the good-set event $\mathcal{T}_{\tilde{\lambda},T}$ and an abstract statistical complexity term $\tilde{\lambda}$. The following theorem, adapted from Feng & Simon (2017), provides a high-probability bound on $\mathcal{T}_{\tilde{\lambda},T}$ for the common cases of regression and classification. This result quantifies the complexity, showing its dependence on the problem dimensions $d$, the number of observations ($n$), and properties of the network.

**Theorem 4.2** (Probabilistic Guarantee for Event $\mathcal{T}_{\tilde{\lambda},T}$)**.** *Consider the following two settings from Feng & Simon (2017):*

    1. **Regression setting:** *Let the loss $\ell(\cdot, \cdot)$ be the squared-error loss, and the data be generated as $Y = f^*(X) + \epsilon$, where $\epsilon$ is a sub-gaussian random variable with mean zero, independent of X.*

2. **Classification setting:** *Let the loss $\ell(\cdot, \cdot)$ be the logistic loss, and the data be generated as $Y \in \{0, 1\}$ where $P(Y = 1|X) = f^*(X)$.*

*Suppose that the set of optimal neural networks $EQ^*$ is composed of $Q$ equivalence classes. For each setting, there exists a constant $c_0 > 0$ such that for*

$$\tilde{\lambda} = c_0 M_1 (d_1 + K\sqrt{d_{upper}}) \sqrt{\frac{\log n}{n}} \left( \sqrt{\log Q} + \frac{X_{max}}{c_1} \log(nc_2) \sqrt{\log(c_2 d)} \right), \tag{11}$$

*we have for any $T \geq 1$,*

$$Pr_{X,Y}(\mathcal{T}_{\tilde{\lambda}, T}) \geq 1 - O\left(\frac{1}{n}\right)$$

*where $c_1 = 1/\sqrt{d_1}$, $c_2 = d_1 + K\sqrt{d_{upper}} + X_{max}/c_1$, and $M_1$ is a constant that bounds the gradient of the loss with respect to the upper-layer parameters.*

This result provides a concrete convergence rate for Theorem 4.1. The statistical complexity $\tilde{\lambda}$ scales roughly as $\sqrt{\log(d)/n}$, and setting the tuning parameter $\lambda \asymp \sqrt{\log(d)/n}$ leads to the estimation error, excess risk and the norm of irrelvant weights converges on the order of $O_p(\sqrt{\log(d)/n} s^2 d_1^{3/2} G)$, $O_p(\log(d)/n s^{5/2} d_1^{3/2} G)$, $O_p(\sqrt{\log(d)/n} s^{5/2} d_1^{3/2} G)$, respectively. These rates demonstrate that the estimator is statistically efficient in high-dimensional, nonparametric settings.

## 4.3 Oracle Properties

Finally, we verify that our estimator possesses the oracle property, a gold standard for variable selection methods. An oracle estimator is an idealized benchmark that knows the true support set $S$ in advance and is defined as the unpenalized minimizer of the empirical loss using only those true variables:

$$\hat{\mathbf{w}}_S^{\mathcal{O}} := \arg\min_{\mathbf{w}_S \in \Theta_S} \left\{ \frac{1}{n} \sum_{i=1}^{n} \ell(f_{\mathbf{w}_S}(X_i), Y_i) \right\},$$

where $\Theta_S$ denotes the subspace of $\Theta$ restricted to the dimensions corresponding to the true model architecture, $\Theta_S = \{\mathbf{w} \in \Theta : \text{supp}(\mathbf{w}) \subseteq S_{param}\}$.

The following theorem shows that, under a minimum signal strength condition, our penalized estimator has a stationary point identical to this ideal oracle estimator, when augmented with zeros for the irrelevant weights. Establishing this result demonstrates that our penalty is capable of correctly distinguishing true signals from noise, effectively mimicking the oracle's perfect knowledge.

**Theorem 4.3** (Existence of a Stationary Point with Oracle Properties)**.** *Suppose the conditions of Theorem 4.1 hold. Assume further that the true nonzero weight groups satisfy the minimum signal strength condition*

$$\min_{j \in S} \|\mathbf{W}_{0,j}^*\|_2 > \delta\lambda + C_0^2(\lambda + T\tilde{\lambda}^*)\sqrt{1 + s},$$

*where $\tilde{\lambda}^*$ denotes the statistical complexity of the oracle model, obtained by replacing the full parameter dimension $d$ with the oracle dimension $d^*$ in the expression for $\tilde{\lambda}$ from Theorem 4.2.*

*Then, on the event $\mathcal{T}_{\tilde{\lambda}^*, T}$, if the regularization parameter satisfies $\lambda \geq C_5 \sqrt{\log(d)/n}$ for some sufficiently large constant $C_5$, the augmented oracle estimator*

$$\hat{\mathbf{w}}^{\mathcal{O}} = \left( \hat{\mathbf{w}}_S^{\mathcal{O}}, \mathbf{0} \right)$$

*is a stationary point of the penalized program equation 4. This holds with probability at least $1 - O(\exp(-C \log d))$ for some constant $C > 0$.*

The proof of Theorem 4.3 is included in B.3. The oracle property provides a strong guarantee of variable selection consistency for our method. Specifically, it implies that when signals are sufficiently strong, the

group concave penalty shrinks the coefficients of irrelevant variables exactly to zero, while leaving the coefficients of relevant variables essentially unbiased, as if they were unpenalized in an oracle estimator. This minimum signal condition is standard for such oracle results in high-dimensional theory; the required signal strength scales with the order of $\sqrt{s\log(d)/n}$ and thus vanishes as the sample size $n$ grows, making the condition weaker for larger datasets. This result offers a rigorous theoretical justification for the method's ability to accurately recover the true sparse model.

## 5 Simulation Studies

We assess the performance of the proposed regularized neural networks in feature selection and prediction through several simulation settings with various types of outcomes. In particular, we consider the concave regularization GMCP and GSCAD for our proposed framework. We name the method of regularized neural networks using GLASSO, GMCP, and GSCAD as GLASSONet, GMCPNet, and GSCADNet, respectively. We compare the proposed group concave regularized estimator GMCPNet and GSCADNet with GLASSONet, neural network (NN) without feature selection ($\lambda = 0$), random (survival) forest (RF), and the STG method proposed in Yamada et al. (2020). These competitors were chosen to provide a comprehensive evaluation: GLASSONet serves as the most direct alternative using a standard convex penalty, NN highlights the challenge of overfitting without sparsity, STG represents a recent state-of-the-art method with a different sparsity mechanism, and RF provides a strong benchmark from the class of tree-based ensembles. We also include the oracle version of NN and RF (Oracle-NN and Oracle-RF) as benchmarks, which are trained with prior knowledge of the true relevant variables, serve as ideal benchmarks for the best achievable performance.

In our implementation, we replace the gradient update in Step (2) with Adam to improve computational efficiency and achieve faster convergence in practice. Specifically, we set the scaling factor $\gamma = 1$ in the thresholding operator and used a base learning rate of LR $= 0.001$ for Adam. For a fair comparison across all neural network methods, we used a ReLU-activated multi-layer perceptron (MLP) with two hidden layers of 10 and 5 units, respectively. As discussed in Section 4.2, ReLU does not strictly satisfy the bounded third-derivative condition of Condition 4.3, since it is non-differentiable at the origin. We used ReLU in our simulations due to its computational efficiency and strong empirical performance. The results obtained with ReLU networks, presented in this section, closely align with the theoretical guarantees, as the non-differentiable points have measure zero for continuous input distributions, and ReLU can be viewed as the limit of smooth activations (e.g., Softplus) for which the condition formally holds.

A sensitivity analysis of the hyperparameter choices, such as network structure and learning rate, is provided in Section 5.3, and additional implementation details can be found in Appendix F.

### 5.1 Preliminary Simulation Study

In this section, we consider regression models of XOR-type and hierarchical signal structures for continuous outcomes. We generate 500 i.i.d. random training samples according to the following models:

- **XOR-type Signals:** $Y = X_1 X_2 + \epsilon$,

- **Hierarchical Signals:** $Y = X_1 + X_1 X_2 + \epsilon$.

In both models, $X_1$ and $X_2$ represent the first two coordinates of the covariate vector $X \in \mathbb{R}^d$, each taking values in $\{\pm 1\}$ with equal probability, while $\epsilon$ denotes a standard normal error term. The remaining coordinates, $X_i$ for $i = 3, \ldots, d$, are uninformative variables. We conducted 20 simulations for varying $d$ values ranging from 20 to 500. For each simulation, the performance of the trained model in both prediction and variable selection was evaluated on 500 independently generated random samples. For prediction accuracy, we report the test $R^2$ scores. For variable selection, we report the false positive rate (FPR)—the percentage of selected but unimportant covariates, defined as FPR $= \frac{|\hat{S} \cap S^c|}{|S^c|} \times 100\%$; and the false negative rate (FNR)—the percentage of important but non-selected covariates, defined as FNR $= \frac{|\hat{S}^c \cap S|}{s} \times 100\%$. Recall that $S$ represents the true index sets of important variables and $\hat{S} = \{j : \|\hat{\mathbf{W}}_{0,j}\|_2 \neq 0\}$ denote the index sets of selected variables. Simulation results are shown in Figure 2.

Our proposed methods, GMCPNet and GSCADNet, demonstrate clear advantages over other approaches, achieving relatively high prediction scores with low FPR and FNR across both models. Importantly, neural networks without feature selection tend to overfit as the number of noisy features increases, underscoring the importance of effective feature selection. Additionally, GLASSONet exhibits a tendency to overselect variables, resulting in a high FPR due to the bias introduced by the LASSO penalty.

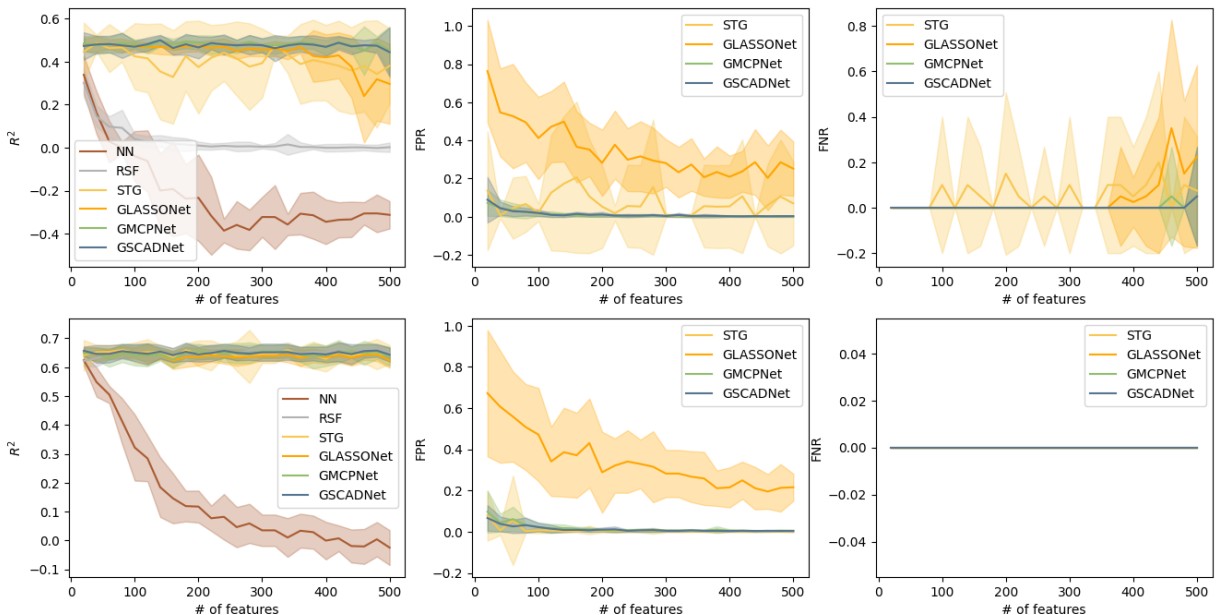

Figure 2: **Top row**: simulation results for the model of XOR-type signals. **Bottom row**: simulation results for the model of hierarchical signals. The $R^2$ scores, false positive rate (FPR), and false negative rate (FNR) are presented in the left, middle, and right columns, respectively. The central lines are the means, while the shaded areas represent standard deviations.

### 5.2 High-Dimensional Simulations with Continuous, Binary, and Time-to-Event Outcomes

We evaluate our proposed methods under a more complex nonlinear pattern across various outcome types, considering both low- and high-dimensional scenarios. The data are generated through the following function:

$$f(X) = \log(|X_1| + 0.1) + X_1 X_2 + X_2 + \exp(X_3 + X_4),$$

where each component of the covariate vector $X = (X_1, \cdots, X_d)^T \in \mathbb{R}^d$ are generated from independent standard normal distribution. Here $d > 4$ and function $f(X)$ is sparse that only the first four variables are relevant to the outcome. We generate $n$ i.i.d. random samples with continuous outcomes, binary outcomes, and time-to-event outcomes in the following three examples, respectively.

**Example 5.1. (Regression Model)** *The continuous response $Y$ is generated from a standard regression model with an additive error as follows*

$$Y = f(X) + \epsilon,$$

*where $\epsilon$ follows a standard normal distribution.*

**Example 5.2. (Classification Model)** *The binary response $Y \in \{0, 1\}$ is generated from a Bernoulli distribution with the following conditional probability*

$$P(Y = 1|X) = \frac{1}{1 + \exp(-f(X))}.$$

**Example 5.3. (Proportional Hazards Model)** *The survival time $T$ follows the proportional hazards model (PHM) with a hazard function*

$$h(t|X) = h_0(t) \exp\left(f(X)\right), \tag{12}$$

*where $h_0(t)$ is the baseline hazard function. Thus, $T = H_0^{-1}\left(-\log(U)\exp f(X)\right)$, where $U$ is a uniform random variable in $[0, 1]$, and $H_0$ is the baseline cumulative hazard function defined as $H_0(t) = \int_0^t h_0(u)du$. We considered a Weibull hazard function for $H_0$, with the scale parameter $= 2$ and the shape parameter $= 2$. A proportion $\mathcal{C}$ of the $n$ samples is randomly selected to be censored. The censoring indicator is defined as $\delta_i = 1$ for observed events and $\delta_i = 0$ for censored observations. The observed time for the ith individual is*

$$Y_i = T_i \mathbb{I}(\delta_i = 1) + C_i \mathbb{I}(\delta_i = 0),$$

*where $T_i$ is the event time and $C_i$ is the censoring time. For censored individuals, $C_i$ is drawn from an independent uniform distribution $(0, T_i)$, ensuring that censoring precedes the event. In our simulation studies, we consider censoring proportions $\mathcal{C} = 0$, 0.2, and 0.4.*

For each example, we consider the low and high dimensional settings in the following scenarios:

1. Low dimension (LD): $d = 20$ and $n = 300$ and $500$.

2. High dimension (HD): $d = 1000$ and $n = 500$.

We perform 200 simulations for each scenario. Similar to Section 5.1, the performance of the trained model is evaluated on independently generated $n$ random samples. For prediction accuracy, we report the $R^2$ score, classification accuracy, and C-index for the regression, classification, and survival models, respectively. In addition to FPR and FNR, we also report the model size (MS), which is the average number of selected covariates.

### 5.2.1 Results

Table 1 presents a summary of the feature selection performance of the four approaches: STG, GLASSONet, GMCPNet, and GSCADNet, across all simulation scenarios. We exclude the results of the STG method for Example 5.3 as it either selects all variables or none of them for the survival outcome. For both LD and HD settings, GMCPNet and GSCADNet consistently outperform the STG and GLASSONet in terms of feature selection. These models exhibit superior performance, achieving model sizes that closely matched the true model, along with low FPR and FNR for most scenarios. While STG performs well in certain LD settings, it tends to over select variables in HD scenarios with a large variability in the model size. On the other hand, GLASSONet is prone to selecting more variables, leading to larger model sizes in both LD and HD settings, which aligns with the inherent nature of the LASSO penalty.

Figure 3 displays the distribution of testing prediction scores for the regression, classification, and PHM with a censoring rate of $\mathcal{C} = 0.2$. The complete results of the PHM are presented in Appendix C. GMCPNet and GSCADNet demonstrate comparable performance in both LD and HD settings, achieving similar results to the Oracle-NN and outperforming NN, RF, and even Oracle-RF in most scenarios. This strong empirical evidence, in which our proposed estimators perform on par with the Oracle-NN possessing prior knowledge of the true features, aligns with the oracle properties established in our theoretical analysis (Theorem 4.3). STG performs similarly to Oracle-NN in the LD setting of the regression model, but its performance deteriorates in the HD setting and other models. Conversely, while GLASSONet outperforms or is comparable to the Oracle-RF method in the LD settings, it suffers from overfitting in the HD settings by including a large number of false positives in the final model.

It is worth pointing out that the Oracle-NN outperforms the Oracle-RF in every scenario, indicating that neural network-based methods can serve as a viable alternative to tree-based methods when the sample size is sufficiently large relative to the number of predictors.

Overall, the simulation results demonstrate the superior performance of the concave penalty in terms of feature selection and prediction. The proposed GMCPNet and GSCADNet methods exhibit remarkable

capabilities in selecting important variables with low FPR and low FNR, while achieving accurate predictions across various models. These methods show promise for tackling the challenges of feature selection and prediction in high-dimensional data.

Table 1: **Feature selection results of STG, GLASSONet, GMCPNet, and GSCADNet under the regression, classification, and proportional hazards models.** The false positive rate (FPR %), false negative rate (FNR %), and model size (MS) with standard deviation (SD) in parentheses are displayed.

| Model | Method | $n = 300, d = 20$ | | $n = 500, d = 20$ | | $n = 500, d = 1000$ | |
|---|---|---|---|---|---|---|---|
| | | FPR, FNR | MS (SD) | FPR, FNR | MS (SD) | FPR, FNR | MS (SD) |
| Regression | STG | 7.8, 5.4 | 5.0 (2.0) | 7.2, 2.1 | 5.1 (1.7) | 1.6, 12.1 | 19.2 (28.0) |
| | GLASSONet | 86.7, 4.4 | 17.7 (4.7) | 96.0, 0.6 | 19.3 (2.2) | 24.3, 29.2 | 245.0 (98.7) |
| | GMCPNet | 2.2, 4.5 | 4.2 (1.0) | 2.1, 4.2 | 4.2 (1.0) | 0.0, 5.8 | 4.1 (0.9) |
| | GSCADNet | 2.4, 5.0 | 4.2 (1.1) | 2.0, 3.2 | 4.2 (0.9) | 0.0, 7.1 | 4.1 (1.0) |
| Classification | STG | 25.3, 16.5 | 7.4 (6.9) | 10.1, 11.0 | 5.2 (4.8) | 3.8, 15.6 | 40.9 (183.4) |
| | GLASSONet | 89.2, 1.0 | 18.2 (2.8) | 94.7, 0.2 | 19.1 (2.0) | 16.3, 21.5 | 165.4 (92.9) |
| | GMCPNet | 14.4, 3.9 | 6.2 (3.6) | 9.3, 0.8 | 5.5 (2.6) | 0.3, 16.2 | 6.5 (4.2) |
| | GSCADNet | 11.6, 5.8 | 5.6 (2.9) | 7.0, 1.0 | 5.1 (1.9) | 0.3, 16.8 | 6.8 (5.9) |
| Survival ($\mathcal{C} = 0$) | GLASSONet | 97.2, 0.0 | 19.5 (1.0) | 99.2, 0.0 | 19.9 (0.5) | 18.2, 20.0 | 184.8 (56.2) |
| | GMCPNet | 1.6, 0.4 | 4.2 (0.6) | 0.8, 0.0 | 4.1 (0.4) | 0.0, 1.5 | 4.1 (0.5) |
| | GSCADNet | 1.9, 0.2 | 4.3 (0.6) | 1.2, 0.0 | 4.2 (0.5) | 0.0, 1.6 | 4.1 (0.7) |
| Survival ($\mathcal{C} = 0.2$) | GLASSONet | 98.0, 0.1 | 19.7 (0.8) | 99.6, 0.0 | 19.9 (0.3) | 16.8, 18.0 | 170.6 (49.0) |
| | GMCPNet | 1.9, 0.4 | 4.3 (0.9) | 1.7, 0.0 | 4.3 (1.0) | 0.0, 2.6 | 4.2 (0.9) |
| | GSCADNet | 1.8, 0.2 | 4.3 (0.9) | 1.7, 0.1 | 4.3 (0.8) | 0.0, 3.5 | 4.1 (0.7) |
| Survival ($\mathcal{C} = 0.4$) | GLASSONet | 95.0, 0.0 | 19.2 (1.7) | 98.8, 0.0 | 19.8 (0.5) | 15.2, 19.9 | 154.6 (48.4) |
| | GMCPNet | 5.8, 8.1 | 4.6 (1.5) | 1.2, 0.1 | 4.2 (0.5) | 0.0, 4.2 | 4.1 (1.0) |
| | GSCADNet | 4.8, 7.5 | 4.5 (1.3) | 1.7, 0.0 | 4.3 (0.7) | 0.0, 4.9 | 4.2 (1.0) |

## 5.3 Hyperparameter Sensitivity Analysis

We evaluate the robustness of the proposed method by conducting a sensitivity analysis under a high-dimensional regression setting ($d = 1000, n = 500$), as described in Example 1. The analysis examines the effect of three key hyperparameters: the thresholding scaling factor ($\gamma$), the learning rate (LR) used in the Adam optimizer, and the network structure. Similar to previous examples, 200 simulations are performed, and the average values of $R^2$, FPR, and FNR are reported for each configuration. Since GMCPNet exhibits similar performance to GSCADNet, we focus on reporting the results for GSCADNet. The findings are summarized in Figure 4, which highlights the sensitivity to each hyperparameter individually while keeping the other parameters fixed. The fixed hyperparameter choices used in this study ($\gamma = 1$, LR = 0.001, and network structure $[10, 5]$, i.e., two hidden layers with 10 and 5 nodes, respectively) are marked on the plots for reference.

For $\gamma$, the results show that $\gamma = 1$ and $\gamma = 0.1$ yield similar performance across all metrics, achieving relatively high $R^2$ while maintaining low FPR and FNR. In contrast, smaller values of $\gamma$ (0.001, 0.01) tend to under-select relevant features, as indicated by higher FNR and consequently lower $R^2$ scores. For LR, values of 0.001, 0.01, and 0.1 exhibit stable performance with competitive $R^2$, low FPR, and low FNR. Smaller LR values (0.0001) lead to slower convergence, resulting in lower $R^2$ and higher FNR, while larger LR values (0.1) slightly improve $R^2$ and reduce FNR but also increase FPR. For network structures, more complex architectures such as $[10, 10, 5]$ and $[20, 10, 5]$ provide modest improvements in $R^2$, though they also

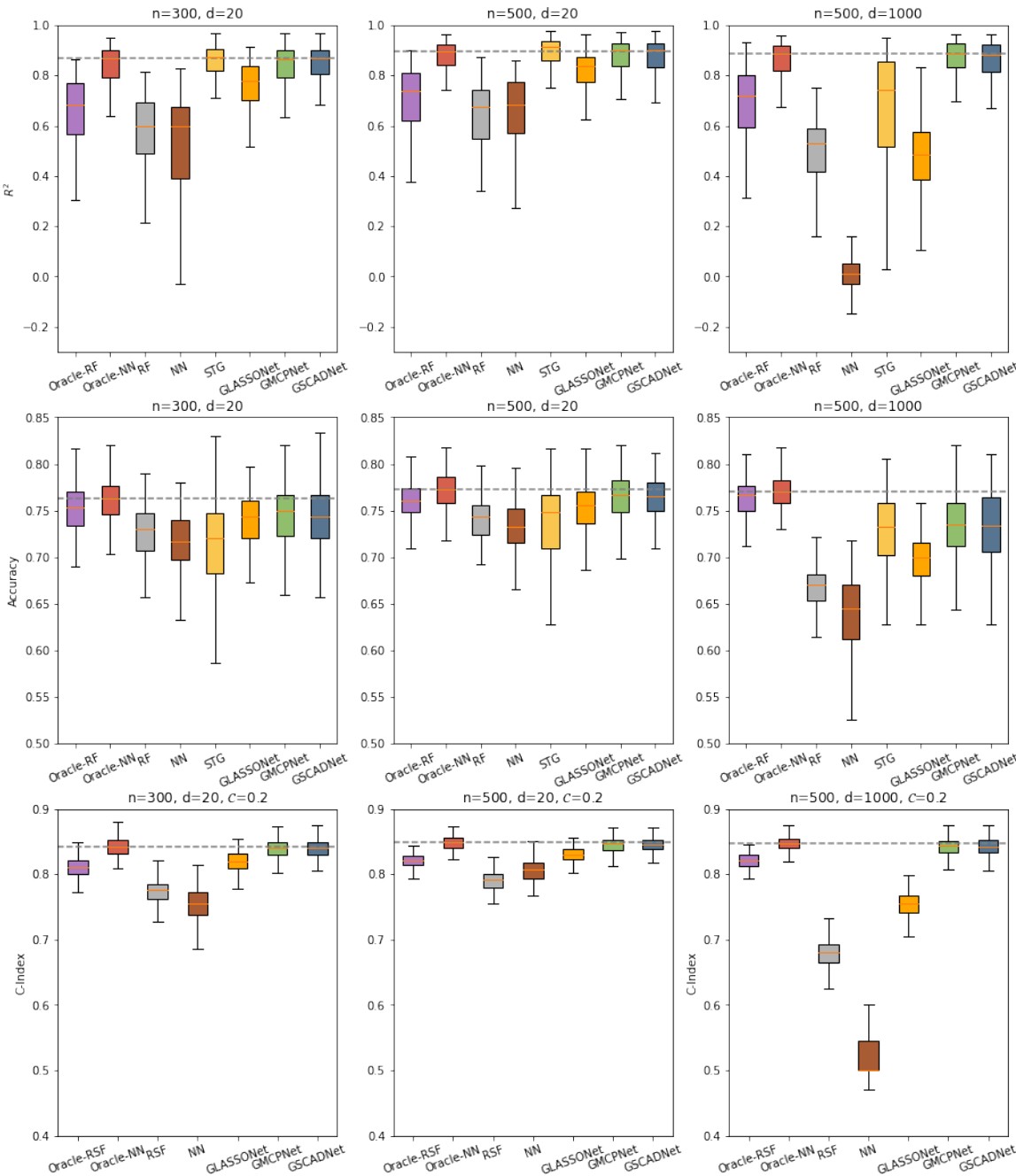

Figure 3: **Top row**: $R^2$ score of the proposed methods for the regression model outlined in Example 5.1. **Middle row**: Accuracy of the proposed methods for the classification model outlined in Example 5.2. **Bottom row**: C-Index of the proposed methods for the survival model outlined in Example 5.3. The dashed lines represent the median score of the Oracle-NN, used as a benchmark for comparison.

result in increased computational costs and potentially a higher risk of overfitting. In summary, although we fixed these parameters in our implementation with LR= 0.001, $\gamma = 1$, and network structure $[10, 5]$), our analysis reveals that selection accuracy and prediction performance remain stable for $\gamma$ between 0.1 and 1, LR between 0.001 and 0.1, and Network structures ranging from $[10,5]$ to more complex architectures like $[20,10,5]$. These findings suggest that our method is robust to a reasonable range of hyperparameter choices, demonstrating consistent performance across different configurations.

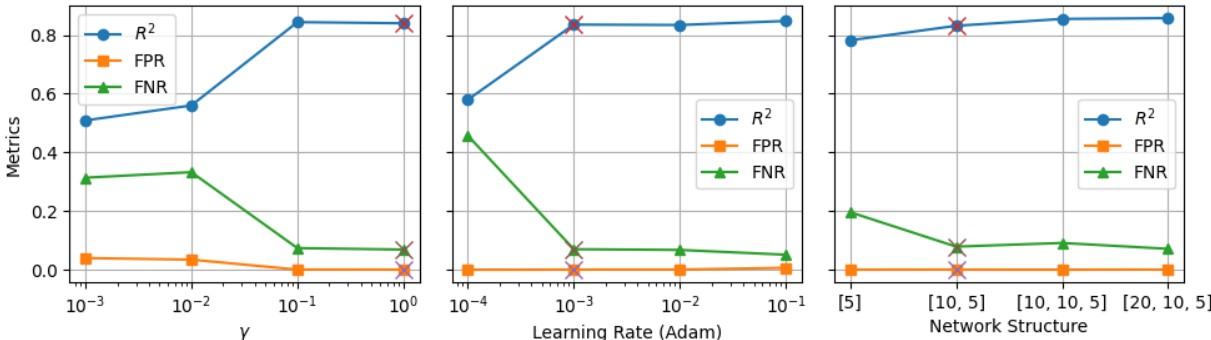

Figure 4: Sensitivity analysis of GSCADNet to hyperparameters: **Left**: $\gamma$, the scaling factor for the thresholding operator. **Middle**: Learning rate (LR) in the Adam optimizer. **Right**: Network structure. The network structure $[l_1, l_2, \ldots, l_k]$ represents the number of nodes in each hidden layer. The fixed choices used in our numerical study ($\gamma = 1$, LR = 0.001, and network structure $[10,5]$) are marked on the plots with an"x" symbol for each metric ($R^2$, FPR, and FNR).

## 6 Real Data Example

In this section, we apply our proposed framework to two real-world datasets to demonstrate its practical utility. The primary goal of these examples is to compare the behavior of different sparse neural network penalties and benchmark our method against strong, standard machine learning approaches, rather than to achieve absolute state-of-the-art performance on these specific tasks. To this end, the CALGB-90401 dataset is used to illustrate the framework's novel application to high-dimensional, time-to-event data, while the MNIST dataset is intentionally subsetted to create a high-dimensional ($p > n$) problem that serves as a visualizable case study for feature selection.

### 6.1 Survival Analysis on CALGB-90401 dataset

We utilize the data from the CALGB-90401 study, a double-blinded phase III clinical trial that compares docetaxel and prednisone with or without bevacizumab in men with metastatic castration-resistant prostate cancer (mCRPC) to illustrate the performance of our proposed method Kelly et al. (2012). The CALGB-90401 GWAS data consists of 498,801 single-nucleotide polymorphisms (SNPs) that are processed from blood samples from patients. We assume a dominant model for SNPs and thus each of the SNPs is considered as a binary variable. Since our interest is studying the DNA damage repair genes, we only consider 625 SNPs based on an updated literature search (Mateo et al., 2015; Wyatt et al., 2016; Mosquera et al., 2013; Robinson et al., 2015; Abida et al., 2019; De Laere et al., 2017; Sumiyoshi et al., 2024). We also include the eight clinical variables that have been identified as prognostic markers of overall survival in patients with mCRPC (Halabi et al., 2014): opioid analgesic use (PAIN), ECOG performance status, albumin (ALB), disease site (defined as lymph node only, bone metastases with no visceral involvement, or any visceral metastases), LDH greater than the upper limit of normal (LDH.High), hemoglobin (HGB), PSA, and alkaline phosphatase (ALKPHOS). The final dataset contains $d = 635$ variables, $n = 631$ patients and a censoring rate $C = 6.8\%$.

We consider the PHM in the form of Eq. (12) for our proposed methods to identify clinical variables or SNPs that can predict the primary outcome of overall survival in these patients. To evaluate the feature selection and prediction performance of the methods, we randomly split the dataset 100 times into training

sets (n=526) and testing sets (n=105) using a 5:1 allocation ratio. We apply the methods to each of the training sets and calculate the time-dependent area under the receiver operating characteristic curve (tAUC) on the corresponding testing sets. The tAUC assesses the discriminative ability of the predicted model and is computed using the Uno method (Uno et al., 2007). The results of the 100 random splits are presented in Figure 5. Our proposed method, GSCADNet, outperforms the others in survival prediction (left panel). It is worth noting that the NN method, which lacks feature selection, tends to overfit in high-dimensional data and performs poorly. Although these three regularized methods of sparse-input neural networks perform similarly in survival prediction, GLASSONet has a tendency to over-select variables and the proposed GMCPNet and GSCADNet select a relatively smaller set of variables without compromising prediction performance (middle panel). The right panel of Figure 5 demonstrates that GSCADNet successfully selects most of the key clinical variables and detects some of the important SNPs in predicting overall survival.

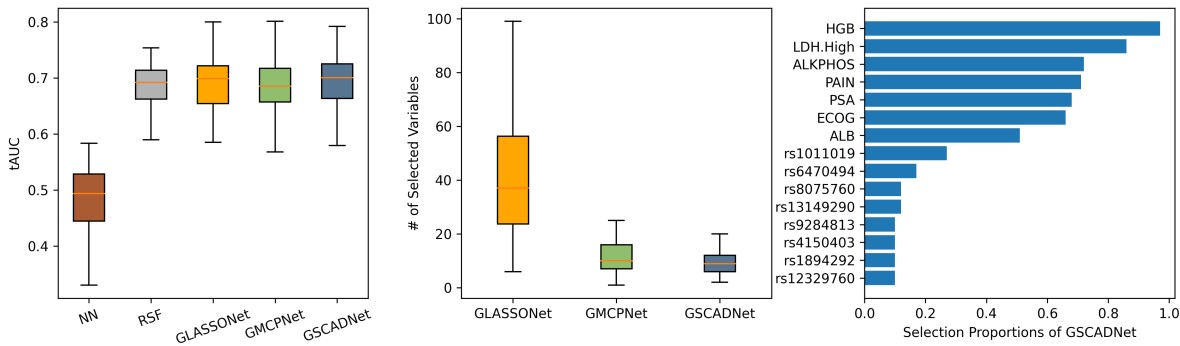

Figure 5: **Left**: Boxplots of tAUC from testing set over 100 random splits. **Middle**: the number of selected variables for GLASSONet, GMCPNet, and GSCADNet. **Right**: Variables selected by GSCADNet with selection proportion$\geq 10\%$ over 100 random splits.

## 6.2 Classification on High-Dimensional MNIST

We aim to visualize variable selection in a high-dimensional binary classification setting using the MNIST dataset. The MNIST dataset is a well-known benchmark dataset in computer vision, consisting of grayscale images of handwritten digits from 0 to 9. In this study, we focus on distinguishing digits 7 and 8, which share structural similarities that make the classification task nontrivial. While other digit pairs may also exhibit visual similarity, this choice provides a meaningful evaluation of feature selection methods in identifying relevant pixels for classification. We evaluate our proposed methods GMCPNet and GSCADNet, along with existing methods GLASSONet, STG, NN, and RF, based on their feature selection and classification accuracy.

The MNIST dataset consists of grayscale images with 28×28 pixels, resulting in 784 variables. To create a high-dimensional, low-sample setting, we construct a training dataset by selecting 250 images of 7s and 8s each, yielding $d = 784$ features and $n = 500$ samples. Importantly, the class labels depend primarily on the central pixels, meaning an effective feature selection method should correctly identify and focus on these relevant regions. To ensure the feature space is not inherently sparse, we introduce i.i.d. standard Gaussian noise to the images. The trained models are evaluated on the testing dataset with 2002 images. We repeated the process of random sampling and model fitting 100 times, and the feature (pixel) selection and classification results are shown in Figure 6. We observe that GLASSONet, GMCPNet, GSCADNet all achieve median accuracies greater than 91%, outperforming the other methods. While the heatmaps of feature selection show that GLASSONet, GMCPNet, GSCADNet consistently select relevant pixels in high frequencies, GLASSONet tends to over select variables and GMCPNet and GSCADNet choose irrelevant pixels in much lower frequencies (indicated by dark red colors).

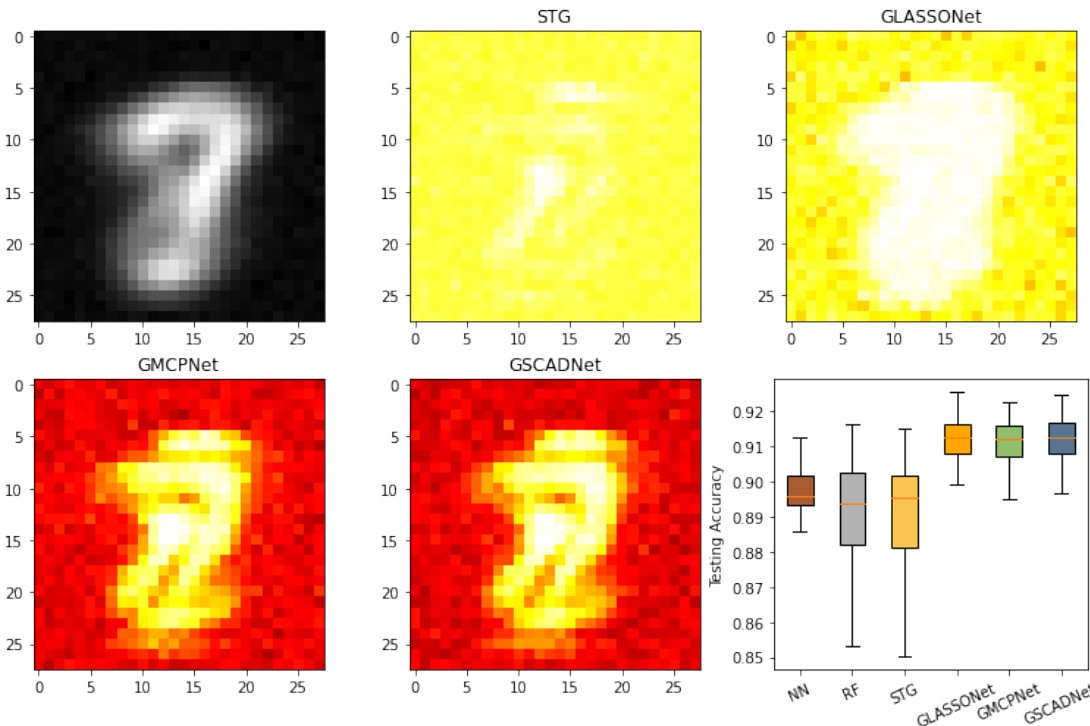

Figure 6: **Comparing feature selection and classification performance by STG, GLASSONet, GMCPNet, and GSCADNet. Top left**: the image that takes the average of all images in the training set and shows relevant pixels in grayscale. **Bottom right**: testing accuracy for the classification of 7s and 8s in a high-dimensional, low-sample MNIST setting with training dataset size $d = 784$ and $n = 500$. **Other panels**: heatmaps depicting the selection frequencies of each pixel across 100 repetitions for each method. Lighter colors indicate higher selection frequencies, with white highest and darker colors lowest.

# 7 Discussion

In this paper, we have proposed a novel framework that utilizes group concave regularization for feature selection and function estimation in complex modeling, specifically designed for sparse-input neural networks. Unlike the convex penalty LASSO, the concave regularization methods such as MCP and SCAD gradually reduce the penalization rate for large terms, preventing over-shrinkage and improving model selection accuracy. Our optimization algorithm, based on the composite gradient descent, is simple to implement, requiring only an additional thresholding operation after the regular gradient descent step on the smooth component. Furthermore, we incorporate backward path-wise optimization to efficiently navigate the optimization landscape across a fine grid of tuning parameters, generating a smooth solution path from dense to sparse models. This path-wise optimization approach improves stability and computational efficiency, potentially enhancing the applicability of our framework for sparse-input neural networks.

Among numerous feature selection methods, penalized regression has been extensively utilized. However, many of these methods rely on the assumption and application of linear theory, which may not capture the complex relationships between covariates and the outcome of interest. In biomedical research, for instance, researchers often normalize data and employ penalized techniques under a linear model for feature selection. However, relying solely on data transformation risks overlooking intricate biological relationships and fails to address the dynamic nature of on-treatment biomarkers. Moreover, advancements in molecular and imaging technologies have introduced challenges in understanding the non-linear relationships between high-dimensional biomarkers and clinical outcomes. Novel approaches are urgently needed to tackle these complexities, leading to an improved understanding of non-linear relationships and optimizing patient treatment and care.

The runtime of our proposed method over a solution path of $\lambda$s (with a fixed $\alpha$) can be comparable to or even shorter than training a single model with a fixed $\lambda$, such as the NN method without feature selection ($\lambda = 0$). To illustrate this, we examine the algorithm complexity of the NN method, which can be approximated as $\mathcal{O}(ndT)$, where $T$ denotes the number of epochs for learning the neural network. In contrast, training our proposed method over a solution path of $m$ $\lambda$s has a complexity of $\mathcal{O}(n\bar{d}T'm)$, where $\bar{d}$ represents the averaged number of inputs along the solution path with dimension pruning, and $T'$ is the number of epochs for each $\lambda$ in the path. In our simulation with the HD scenario ($d = 1000$), we set $T = 5000$, $T' = 200$, and $m = 50$. Assuming the number of inputs decreases equally along the solution path from the full model to the null model, we have $\bar{d} = d/2 = 500$. Thus, $ndT = n\bar{d}T'm$ indicates that solving for an entire path of our proposed method requires a similar computation as training a single model. In real applications, especially in high-dimensional scenarios, the dimensionality usually drops quickly along the solution path. Therefore, $\bar{d}$ can be much smaller than $d/2$, and thus solving for a whole solution path can be more computationally efficient. It is worth pointing out that we set $T'$ to be small for the first parameter $\lambda_{\min}$ as well in the HD setting, to avoid overfitting of an initial dense model.

A central contribution of this work is the rigorous theoretical foundation we establish for the proposed framework. Our analysis provides non-asymptotic, finite-sample guarantees for any suitable local minimizer. In particular, we derive explicit bounds on both estimation error and prediction error (excess risk), showing that the method achieves statistical consistency even in high-dimensional settings. Furthermore, we prove a key oracle property: under regular conditions, the ideal oracle estimator, which is the solution attainable if the true relevant variables were known, is a stationary point of our penalized objective function. Although the non-convexity of neural networks precludes guarantees of convergence to a global minimum, our results demonstrate that the framework is properly formulated to yield statistically reliable estimates and recover the true sparse model. This is further supported by our empirical studies, which indicate that the algorithm tends to converge to solutions consistent with the predicted theoretical performance and often comparable to the Oracle-NN benchmark.

One limitation of the proposed method arises in ultra-high dimensional scenarios where the number of variables reaches hundreds of millions. Directly applying the proposed sparse-input neural networks in such cases can lead to an exceedingly complex optimization landscape, making it computationally infeasible. A potential solution is to employ a pre-screening step to reduce dimensionality before applying the proposed approach (Fan & Lv, 2010).

Another limitation of the proposed group-regularized method is its focus on individual feature selection. This limitation becomes particularly relevant when dealing with covariates exhibiting grouping structures, such as a group of indicator variables representing a multilevel categorical covariate, or scientifically meaningful groups based on prior knowledge. A potential future research direction could involve redefining the groups within the proposed framework. This could be achieved by considering all outgoing connections from a group of input neurons as a single group, enabling group selection and accommodating the presence of grouping structures.

Furthermore, our study is limited by its focus on a standard MLP architecture. This choice was intentional, as it provides a clear setting to illustrate the properties of the proposed feature selection framework without the confounding influence of complex architectural priors. We recognize, however, that this choice may not yield state-of-the-art predictive performance on tasks where specialized architectures, such as convolutional neural networks for image data, are more effective. A natural direction for future research is to integrate our group concave regularization approach with such architectures, potentially achieving models that combine strong predictive accuracy with the feature-level interpretability offered by our framework.

In conclusion, our study demonstrates the advantages of employing group concave regularization for sparse-input neural networks. Our theoretical results establish finite-sample guarantees, demonstrating statistical consistency and oracle-like performance of the method under regular conditions. The findings highlight its effectiveness in consistently selecting relevant variables and accurately modeling complex non-linear relationships between covariates and outcomes across both low- and high-dimensional settings. The proposed approach holds promising potential to enhance modeling strategies and find wide-ranging applications, particularly in diseases characterized by non-linear biomarkers, such as oncology and infectious diseases.

### Acknowledgments

This research was supported in part by the National Institutes of Health grants R01CA256157, R01CA249279, R21CA263950, the US Food and Drug Administration grant 1U01FD007857-01, the United States Army Medical Research Materiel Command grants HT9425-23-1-0393 and HT9425-25-1-0623, and the Prostate Cancer Foundation Challenge Award.

## A    Empirical Loss Function

The empirical loss functions $\mathcal{L}_n(\mathbf{w})$ for regression, classification, and survival models in Examples 5.1-5.3 are defined as follows:

- Mean squared error loss for regression tasks. This loss function measures the average squared difference between the true values $Y_i$ and the predictions $f_{\mathbf{w}}(X_i)$:

$$\mathcal{L}_n(\mathbf{w}) = \frac{1}{n} \sum_{i=1}^{n} (Y_i - f_{\mathbf{w}}(X_i))^2.$$

- Cross-entropy loss for classification tasks. It is widely used in classification problems and quantifies the dissimilarity between the true labels $Y_i$ and the predicted probabilities $\hat{Y}_i$ of class 1. The predicted probability $\hat{Y}_i$ is obtained by applying the sigmoid function to $f_{\mathbf{w}}(X_i)$:

$$\mathcal{L}_n(\mathbf{w}) = -\frac{1}{n} \sum_{i=1}^{n} \left[ Y_i \log(\hat{Y}_i) + (1 - Y_i) \log(1 - \hat{Y}_i) \right].$$

- Negative log partial likelihood for proportional hazards models. It is derived from survival analysis and aims to maximize the likelihood of observing events while considering censoring information. It incorporates the event indicator $\delta_i$, which is 1 if the event of interest occurs at time $Y_i$ and 0 if the observation is right-censored. The negative log partial likelihood is defined as:

$$\mathcal{L}_n(\mathbf{w}) = -\frac{1}{n} \sum_{i=1}^{n} \left\{ \delta_i f_{\mathbf{w}}(X_i) - \delta_i \log \sum_{j \in R_i} \exp(f_{\mathbf{w}}(X_j)) \right\}.$$

Here, $R_i = \{j : Y_j \geq Y_i\}$ represents the risk set just before time $Y_i$. The negative log partial likelihood is specifically used in the proportional hazards model.

## B Proofs

### B.1 Proofs of Theorem 4.1

**Lemma B.1** (Quadratic lower bound). *Suppose Conditions 4.1, 4.2, and 4.3 hold. For some constant $R > 0$, suppose $\mathbf{w} \in \Theta$ satisfies*

$$||\mathbf{w}_S - \mathbf{w}^{*(\mathbf{w})}||_2 \leq R \quad and \tag{13}$$

$$\sum_{j \in S^c} ||\mathbf{W}_{0,j}||_2 \leq \left( ||\mathbf{w}_{upper} - \mathbf{w}^{*(\mathbf{w})}_{upper}||_2 + 5 \sum_{j \in S} ||\mathbf{W}_{0,j} - \mathbf{W}^{*(\mathbf{w})}_{0,j}||_2 \right). \tag{14}$$

*Then we have $\mathcal{E}(\mathbf{w}) \geq ||\mathbf{w}_S - \mathbf{w}^{*(\mathbf{w})}||_2^2 / C_0^2$ where*

$$C_0^2 = \frac{1}{\epsilon_0} \vee \frac{R^2}{\chi_{\epsilon_0}} \tag{15}$$

$$\epsilon_0 = \frac{h_{min}}{\frac{2}{3} G (\sqrt{d_1}(1 + 5\sqrt{s}) + \sqrt{d^*})^3}. \tag{16}$$

*Proof.* We begin by defining the $l_2$ distance between the relevant parameters of $\mathbf{w}$ and the closest optimal parameters $\mathbf{w}^{*(\mathbf{w})}$ as $D(\mathbf{w}) = ||\mathbf{w}_S - \mathbf{w}^{*(\mathbf{w})}_S||_2$. The objective is to establish a quadratic lower bound for the excess loss $\mathcal{E}(\mathbf{w})$ in terms of $D(\mathbf{w})$.

Since the gradient of the expected loss vanishes at the optimum, a third-order Taylor expansion of the excess loss around $\mathbf{w}^{*(\mathbf{w})}$ yields:

$$\mathcal{E}(\mathbf{w}) = \frac{1}{2}(\mathbf{w} - \mathbf{w}^{*(\mathbf{w})})^\top [\nabla^2_{\mathbf{w}} \mathbb{E}\ell(f_{\mathbf{w}}(X), Y)]_{\mathbf{w}=\mathbf{w}^{*(\mathbf{w})}} (\mathbf{w} - \mathbf{w}^{*(\mathbf{w})}) + r_{\mathbf{w}} \tag{17}$$

where $r_{\mathbf{w}}$ is the Lagrange remainder term.

From Condition 4.1, the quadratic term is lower-bounded by:

$$(\mathbf{w} - \mathbf{w}^{*(\mathbf{w})})^\top [\nabla^2_{\mathbf{w}} \mathbb{E}\ell(f_{\mathbf{w}}(X), Y)]_{\mathbf{w}=\mathbf{w}^{*(\mathbf{w})}} (\mathbf{w} - \mathbf{w}^{*(\mathbf{w})}) \geq h_{min} D^2(\mathbf{w}). \tag{18}$$

From Condition 4.3, the remainder term is bounded as:

$$|r_{\mathbf{w}}| \leq \frac{G}{6} ||\mathbf{w} - \mathbf{w}^{*(\mathbf{w})}||_1^3 \tag{19}$$

The subsequent step is to bound the $l_1$-norm $||\mathbf{w} - \mathbf{w}^{*(\mathbf{w})}||_1$ in terms of $D(\mathbf{w})$. We decompose the norm into components corresponding to the irrelevant weights ($S^c$) and the relevant parameters ($\mathbf{w}_S$):

$$||\mathbf{w} - \mathbf{w}^{*(\mathbf{w})}||_1 = \sum_{j \in S^c} ||\mathbf{W}_{0,j}||_1 + ||\mathbf{w}_S - \mathbf{w}^{*(\mathbf{w})}_S||_1 \tag{20}$$

noting that $\mathbf{W}^{*(\mathbf{w})}_{0,j} = \mathbf{0}$ for all $j \in S^c$. The two terms of this decomposition are bounded separately.

The first term, corresponding to the irrelevant weights, is bounded using the Cauchy-Schwarz inequality, $||\mathbf{W}_{0,j}||_1 \leq \sqrt{d_1}||\mathbf{W}_{0,j}||_2$, which leads to:

$$\sum_{j \in S^c} ||\mathbf{W}_{0,j}||_1 \leq \sqrt{d_1} \sum_{j \in S^c} ||\mathbf{W}_{0,j}||_2$$

Under (14) assumed in the lemma, the sum of the $l_2$ norms is bounded, yielding:

$$\sum_{j \in S^c} ||\mathbf{W}_{0,j}||_1 \leq \sqrt{d_1} \left( ||\mathbf{w}_{upper} - \mathbf{w}_{upper}^{*(\mathbf{w})}||_2 + 5 \sum_{j \in S} ||\mathbf{W}_{0,j} - \mathbf{W}_{0,j}^{*(\mathbf{w})}||_2 \right)$$

$$\leq \sqrt{d_1} \left( ||\mathbf{w}_{upper} - \mathbf{w}_{upper}^{*(\mathbf{w})}||_2 + 5\sqrt{s}||\mathbf{W}_{0,S} - \mathbf{W}_{0,S}^{*(\mathbf{w})}||_2 \right)$$

$$\leq \sqrt{d_1}(1 + 5\sqrt{s})D(\mathbf{w})$$

The second term, for the relevant parameters, is bounded similarly. Let $d^*$ be the dimension of $\mathbf{w}_S$. By Cauchy-Schwarz:

$$||\mathbf{w}_S - \mathbf{w}_S^{*(\mathbf{w})}||_1 \leq \sqrt{d^*}||\mathbf{w}_S - \mathbf{w}_S^{*(\mathbf{w})}||_2 = \sqrt{d^*}D(\mathbf{w})$$

Summing the bounds for these two components provides an upper bound for the total $l_1$ norm:

$$||\mathbf{w} - \mathbf{w}^{*(\mathbf{w})}||_1 \leq \sqrt{d_1}(1 + 5\sqrt{s})D(\mathbf{w}) + \sqrt{d^*}D(\mathbf{w})$$

$$= \left( \sqrt{d_1}(1 + 5\sqrt{s}) + \sqrt{d^*} \right) D(\mathbf{w})$$

Substituting this into the remainder inequality equation 19 gives:

$$|r_{\mathbf{w}}| \leq \frac{G}{6} \left( \sqrt{d_1}(1 + 5\sqrt{s}) + \sqrt{d^*} \right)^3 D^3(\mathbf{w}) \tag{21}$$

By combining the lower bound on the quadratic term (18) with the upper bound for the magnitude of the remainder (21), we establish the following lower bound for the excess risk:

$$\mathcal{E}(\mathbf{w}) \geq \frac{1}{2} h_{min} D^2(\mathbf{w}) - \frac{G}{6} \left( \sqrt{d_1}(1 + 5\sqrt{s}) + \sqrt{d^*} \right)^3 D^3(\mathbf{w})$$

Now we use Condition 4.2 and apply Auxiliary Lemma Städler et al. (2010), as it is of the form $h(z) \geq \Lambda^2 z^2 - C z^3$ with $z = D(\mathbf{w})$. Therefore, applying the lemma yields the desired result:

$$\mathcal{E}(\mathbf{w}) \geq D^2(\mathbf{w})/C_0^2$$

where

$$C_0^2 = \frac{1}{\epsilon_0} \vee \frac{R^2}{\chi_{\epsilon_0}} \quad \text{and}$$

$$\epsilon_0 = \frac{h_{min}}{\frac{2}{3}G(\sqrt{d_1}(1 + 5\sqrt{s}) + \sqrt{d^*})^3}.$$

$\square$

**Lemma B.2** (Lower Bound for a Class of Nonconvex Penalties). *Let the penalty function $\rho_\lambda(t)$ satisfy Condition 4.4. Then for any $t \in [0, \delta\lambda]$, the penalty satisfies the lower bound:*

$$\rho_\lambda(t) \geq \frac{1}{2}\lambda t$$

*Proof.* By Condition 4.4-(iii), the function $g(t) = \rho_\lambda(t)/t$ is non-increasing. Thus, for any $t \in (0, \delta\lambda]$, $g(t) \geq g(\delta\lambda)$, which implies $\rho_\lambda(t)/t \geq \rho_\lambda(\delta\lambda)/\delta\lambda$. The proof reduces to showing $\rho_\lambda(\delta\lambda)/\delta\lambda \geq \lambda/2$.

By Conditions 4.4-(v), (vi), and (vii), the derivative $\rho_\lambda'(u)$ is a concave function on $[0, \delta\lambda]$ with $\rho_\lambda'(0) = \lambda$ and $\rho_\lambda'(\delta\lambda) = 0$. A concave function's graph lies above its secant line. The secant line connecting $(0, \lambda)$ and $(\delta\lambda, 0)$ is $L(u) = \lambda(1 - u/\delta\lambda)$. Therefore, $\rho_\lambda'(u) \geq L(u)$ for $u \in [0, \delta\lambda]$.

Integrating both sides from 0 to $\delta\lambda$ gives:

$$\rho_\lambda(\delta\lambda) = \int_0^{\delta\lambda} \rho'_\lambda(u)\,du \geq \int_0^{\delta\lambda} L(u)\,du = \frac{1}{2}\lambda\delta\lambda$$

Dividing by $\delta\lambda$ yields $\rho_\lambda(\delta\lambda)/\delta\lambda \geq \lambda/2$, which completes the proof. $\qquad\square$

Now we are ready to prove Theorem 4.1. We first suppose the existence of a local minimizer $\hat{\mathbf{w}}$ that satisfies $\max_{j\in S^c} \|\hat{\mathbf{W}}_{0,j}\|_2 \leq \delta\lambda$; we will establish this fact at the end of the proof. Then the proof proceeds in three main steps. First, we establish a basic inequality derived from the optimality of $\hat{\mathbf{w}}$. Second, we use this inequality to verify that the compatibility condition holds. Finally, by combining the basic inequality with the quadratic lower bound from Lemma B.1, we derive the final bounds on estimation error, excess risk, and sparsity.

For simplicity, we denote $\mathbf{w}^{*(\hat{\mathbf{w}})} = \mathbf{w}^*$.

**Step 1: Basic Inequality.** By definition of $\hat{\mathbf{w}}$ as a minimizer and using Condition 4.4-(i) ($\rho_\lambda(0) = 0$):

$$\frac{1}{n}\sum_{i=1}^n \ell_{\hat{\mathbf{w}}}(x_i, y_i) + \sum_{j=1}^d \rho_\lambda(\|\hat{\mathbf{W}}_{0,j}\|_2) \leq \frac{1}{n}\sum_{i=1}^n \ell_{\mathbf{w}^*}(x_i, y_i) + \sum_{j\in S} \rho_\lambda(\|\mathbf{W}^*_{0,j}\|_2).$$

Rearranging and using the definition of excess loss gives:

$$\mathcal{E}(\hat{\mathbf{w}}) + \sum_{j=1}^d \rho_\lambda(\|\hat{\mathbf{W}}_{0,j}\|_2) \leq (\mathbb{E}_n - \mathbb{E})(\ell_{\mathbf{w}^*} - \ell_{\hat{\mathbf{w}}}) + \sum_{j\in S} \rho_\lambda(\|\mathbf{W}^*_{0,j}\|_2).$$

Over the event $\mathcal{T}_{\tilde{\lambda},T}$, let $\Delta(\hat{\mathbf{w}}, \mathbf{w}^*) = \|\hat{\mathbf{w}}_{upper} - \mathbf{w}^*_{upper}\|_2 + \sum_{j=1}^d \|\hat{\mathbf{W}}_{0,j} - \mathbf{W}^*_{0,j}\|_2$.

The proof now proceeds by considering two cases based on the magnitude of $\Delta(\hat{\mathbf{w}}, \mathbf{w}^*)$.

**Case 1:** $\Delta(\hat{\mathbf{w}}, \mathbf{w}^*) \leq \tilde{\lambda}$.

In this scenario, the total estimation error $\Delta(\hat{\mathbf{w}}, \mathbf{w}^*)$ is already bounded by the statistical complexity term $\tilde{\lambda}$. The conclusions of the theorem are thus met, as the claimed upper bounds are of order $\lambda$ or $\tilde{\lambda}$ (or higher), which will be larger than the assumed error for any non-trivial choice of regularization.

To be more explicit, the individual bounds are satisfied as follows:

(i) **Estimation Error:** The quantity to be bounded is $\|\hat{\mathbf{w}}_S - \mathbf{w}^*_S\|_2$. By definition, $\|\hat{\mathbf{w}}_S - \mathbf{w}^*_S\|_2 \leq \Delta(\hat{\mathbf{w}}, \mathbf{w}^*) \leq \tilde{\lambda}$. The theorem's bound of $C_0^2(\lambda + T\tilde{\lambda})\sqrt{1+s}$ is of order $\lambda$ or $\tilde{\lambda}$, and thus the inequality holds.

(ii) **Excess Risk:** We start from the basic inequality derived from the optimality of $\hat{\mathbf{w}}$:

$$\mathcal{E}(\hat{\mathbf{w}}) + \sum_{j\in S^c} \rho_\lambda(\|\hat{\mathbf{W}}_{0,j}\|_2) \leq (\mathbb{E}_n - \mathbb{E})(\ell_{\mathbf{w}^*} - \ell_{\hat{\mathbf{w}}}) + \sum_{j\in S} \left(\rho_\lambda(\|\mathbf{W}^*_{0,j}\|_2) - \rho_\lambda(\|\hat{\mathbf{W}}_{0,j}\|_2)\right).$$

We bound the terms on the right-hand side. First, the stochastic error term is bounded by the definition of the event $\mathcal{T}_{\tilde{\lambda},T}$:

$$(\mathbb{E}_n - \mathbb{E})(\ell_{\mathbf{w}^*} - \ell_{\hat{\mathbf{w}}}) \leq T\tilde{\lambda}(\tilde{\lambda} \vee \Delta).$$

In this case, since $\Delta \leq \tilde{\lambda}$, this term is bounded by $T\tilde{\lambda}^2$.

Next, because the penalty function $\rho_\lambda$ is Lipschitz with constant $\lambda$ (from Conditions 4.4-(iii,v)), the second term is bounded by:

$$\sum_{j\in S} \left(\rho_\lambda(\|\mathbf{W}^*_{0,j}\|_2) - \rho_\lambda(\|\hat{\mathbf{W}}_{0,j}\|_2)\right) \leq \lambda \sum_{j\in S} \|\mathbf{W}^*_{0,j} - \hat{\mathbf{W}}_{0,j}\|_2 \leq \lambda\Delta.$$

Again, since $\Delta \leq \tilde{\lambda}$, this term is bounded by $\lambda\tilde{\lambda}$.

Combining these bounds and noting that the penalty on the inactive set $S^c$ is non-negative, we can drop it to get:

$$\mathcal{E}(\hat{\mathbf{w}}) \leq T\tilde{\lambda}^2 + \lambda\tilde{\lambda} = (T\tilde{\lambda} + \lambda)\tilde{\lambda}.$$

This derived upper bound is of order $\tilde{\lambda}^2$ or $\lambda\tilde{\lambda}$. This is consistent with and smaller than the final bound of $(\lambda + T\tilde{\lambda})^2(1+s)C_0^2$ stated in the theorem, which is derived from the more critical second case. Thus, the conclusion for the excess risk holds.

(iii) **Sparsity Recovery:** The quantity to be bounded is $\sum_{j \in S^c} ||\hat{\mathbf{W}}_{0,j}||_2$. By definition, this is also less than or equal to $\Delta(\hat{\mathbf{w}}, \mathbf{w}^*) \leq \tilde{\lambda}$. The theorem's bound is of order $\lambda$ or $\tilde{\lambda}$, so this condition is also satisfied.

**Case 2:** $\Delta(\hat{\mathbf{w}}, \mathbf{w}^*) > \tilde{\lambda}$.

We now focus on the more challenging case where the estimation error is large. We start again from the basic inequality:

$$\mathcal{E}(\hat{\mathbf{w}}) + \sum_{j \in S^c} \rho_\lambda(||\hat{\mathbf{W}}_{0,j}||_2) \leq T\tilde{\lambda}\Delta(\hat{\mathbf{w}}, \mathbf{w}^*) + \sum_{j \in S}(\rho_\lambda(||\mathbf{W}_{0,j}^*||_2) - \rho_\lambda(||\hat{\mathbf{W}}_{0,j}||_2)).$$

By Lemma B.2, since $\max_{j \in S^c} ||\hat{\mathbf{W}}_{0,j}||_2 \leq \delta\lambda$, we have $\rho_\lambda(||\hat{\mathbf{W}}_{0,j}||_2) \geq \frac{1}{2}\lambda||\hat{\mathbf{W}}_{0,j}||_2$ for $j \in S^c$. Additionally, from Conditions 4.4-(iii,v), the penalty is Lipschitz with constant $\lambda$, i.e., $|\rho_\lambda(t_1) - \rho_\lambda(t_2)| \leq \lambda|t_1 - t_2|$. This gives:

$$\mathcal{E}(\hat{\mathbf{w}}) + \frac{1}{2}\lambda\sum_{j \in S^c} ||\hat{\mathbf{W}}_{0,j}||_2 \leq T\tilde{\lambda}\left(||\hat{\mathbf{w}}_{upper} - \mathbf{w}_{upper}^*||_2 + \sum_{j \in S^c} ||\hat{\mathbf{W}}_{0,j}||_2\right) + (\lambda + T\tilde{\lambda})\sum_{j \in S} ||\hat{\mathbf{W}}_{0,j} - \mathbf{W}_{0,j}^*||_2.$$

Grouping terms yields:

$$\mathcal{E}(\hat{\mathbf{w}}) + \left(\frac{1}{2}\lambda - T\tilde{\lambda}\right)\sum_{j \in S^c} ||\hat{\mathbf{W}}_{0,j}||_2 \leq T\tilde{\lambda}||\hat{\mathbf{w}}_{upper} - \mathbf{w}_{upper}^*||_2 + (\lambda + T\tilde{\lambda})\sum_{j \in S} ||\hat{\mathbf{W}}_{0,j} - \mathbf{W}_{0,j}^*||_2. \qquad (22)$$

**Step 2: Verifying the Compatibility Condition.** We choose $\lambda$ such that $\lambda \geq 4T\tilde{\lambda}$. This ensures $\frac{1}{2}\lambda - T\tilde{\lambda} \geq 2T\tilde{\lambda} - T\tilde{\lambda} = T\tilde{\lambda} > 0$. From equation 22:

$$\sum_{j \in S^c} ||\hat{\mathbf{W}}_{0,j}||_2 \leq \frac{T\tilde{\lambda}}{\frac{1}{2}\lambda - T\tilde{\lambda}}||\hat{\mathbf{w}}_{upper} - \mathbf{w}_{upper}^*||_2 + \frac{\lambda + T\tilde{\lambda}}{\frac{1}{2}\lambda - T\tilde{\lambda}}\sum_{j \in S} ||\hat{\mathbf{W}}_{0,j} - \mathbf{W}_{0,j}^*||_2.$$

The first coefficient is $\leq 1$ and the second is $\leq 5$. Thus, Condition 13 is satisfied. In addition, by assumption we have

$$||\hat{\mathbf{w}}_S - \mathbf{w}_S^*||_2 \leq ||\hat{\mathbf{w}}||_2 + ||\mathbf{w}^*||_2 \leq 2K.$$

Therefore, the conditions of Lemma B.1 are satisfied. Applying the lemma yields

$$\mathcal{E}(\hat{\mathbf{w}}) \geq \frac{||\hat{\mathbf{w}}_S - \mathbf{w}_S^*||_2^2}{C_0^2}.$$

**Step 3: Deriving the Final Bound and Proving Existence.** Let $D(\hat{\mathbf{w}}) = ||\hat{\mathbf{w}}_S - \mathbf{w}_S^*||_2$. From equation 22 and Cauchy-Schwarz:

$$\mathcal{E}(\hat{\mathbf{w}}) + \left(\frac{1}{2}\lambda - T\tilde{\lambda}\right)\sum_{j \in S^c} ||\hat{\mathbf{W}}_{0,j}||_2 \leq (\lambda + T\tilde{\lambda})\sqrt{1+s}D(\hat{\mathbf{w}}).$$

Applying Young's inequality to the right-hand side gives $\leq \frac{1}{2}(\lambda + T\tilde{\lambda})^2(1+s)C_0^2 + \frac{D(\hat{\mathbf{w}})^2}{2C_0^2}$. Substituting the lower bound for $\mathcal{E}(\hat{\mathbf{w}})$:

$$\mathcal{E}(\hat{\mathbf{w}}) + \left(\frac{1}{2}\lambda - T\tilde{\lambda}\right) \sum_{j \in S^c} ||\hat{\mathbf{W}}_{0,j}||_2 \leq \frac{1}{2}(\lambda + T\tilde{\lambda})^2(1+s)C_0^2 + \frac{1}{2}\mathcal{E}(\hat{\mathbf{w}}).$$

Multiplying by 2 and rearranging yields the main result:

$$\mathcal{E}(\hat{\mathbf{w}}) + (\lambda - 2T\tilde{\lambda}) \sum_{j \in S^c} ||\hat{\mathbf{W}}_{0,j}||_2 \leq (\lambda + T\tilde{\lambda})^2(1+s)C_0^2.$$

This holds for any local minimizer satisfying our initial assumption. To show such a minimizer exists, we must confirm that the parameter $\delta$ of the penalty function can be chosen to be consistent with this result. From the derived bound, we have:

$$\max_{j \in S^c} ||\hat{\mathbf{W}}_{0,j}||_2 \leq \frac{(\lambda + T\tilde{\lambda})^2(1+s)C_0^2}{\lambda - 2T\tilde{\lambda}}.$$

We require this upper bound to be less than or equal to $\delta\lambda$:

$$\frac{(\lambda + T\tilde{\lambda})^2(1+s)C_0^2}{\lambda - 2T\tilde{\lambda}} \leq \delta\lambda.$$

Let $\lambda = cT\tilde{\lambda}$ for a constant $c \geq 4$. After substitution and canceling $T\tilde{\lambda}$ on both sides, the condition becomes:

$$\frac{T\tilde{\lambda}(c+1)^2(1+s)C_0^2}{c-2} \leq \delta cT\tilde{\lambda} \implies \frac{(c+1)^2(1+s)C_0^2}{c(c-2)} \leq \delta.$$

Since the left-hand side is a finite constant that depends on the problem parameters (but not on $n$ or $d$), we can always choose a penalty function with a parameter $\delta$ large enough to satisfy this condition. This confirms that a local minimizer $\hat{\mathbf{w}}$ satisfying our initial assumption exists for a suitable choice of penalty.

## B.2 Proof of Theorem 4.2

The proof of this theorem is a direct adaptation of the proof of Theorem 2 in Feng & Simon (2017). It relies on empirical process theory to bound the stochastic error term.

The core logic follows their use of entropy bounds for neural network function classes (Lemma 2 in their work) and a peeling argument (Lemma 4 in their work). The primary modification for our setting involves adapting their bounds from the sparse group LASSO penalty to the group LASSO penalty used here. This corresponds to setting their parameter $\alpha = 1$ in the relevant constants ($c_1$ and $c_2$), which leads to the formula for $\tilde{\lambda}$ presented in the theorem statement. We refer the interested reader to the original paper for the complete technical details.

## B.3 Proof of Theorem 4.3

This section provides a complete proof of Theorem 4.3, which establishes that the oracle estimator is a stationary point of the penalized problem. The proof follows the Primal-Dual-Witness (PDW) method, adapted for the non-convex setting of sparse neural networks.

## B.4 Foundational Lemmas

First, we establish a crucial lemma that provides a tighter estimation error bound for the oracle estimator and shows that it lies in the interior of the feasible set $\Theta$ with high probability. This allows us to use the simpler zero-gradient condition for optimality.

**Lemma B.3** (Oracle and Restricted Estimator Properties and Equivalence). *Suppose the conditions of Theorem 4.1 are met. Let an oracle estimator $\hat{\mathbf{w}}_S^{\mathcal{O}}$ and a restricted penalized estimator $\tilde{\mathbf{w}}_S$ be defined as any solution to their respective programs:*

$$\hat{\mathbf{w}}_S^{\mathcal{O}} \in \arg\min_{\mathbf{w}_S \in \Theta_S} \left\{ \frac{1}{n} \sum_{i=1}^n \ell(f_{\mathbf{w}_S}(X_i), Y_i) \right\}$$

$$\tilde{\mathbf{w}}_S \in \arg\min_{\mathbf{w}_S \in \Theta_S} \left\{ \frac{1}{n} \sum_{i=1}^n \ell(f_{\mathbf{w}_S}(X_i), Y_i) + \sum_{j \in S} \rho_\lambda(\|\mathbf{W}_{0,j}\|_2) \right\}$$

*where $\Theta_S$ is the parameter space restricted to the active set $S$. Then, with probability at least $1 - O(1/n)$:*

(a) **Estimation Error:** *The estimators satisfy the following bounds:*

   (i) $\|\hat{\mathbf{w}}_S^{\mathcal{O}} - \mathbf{w}_S^*\|_2 \le C_0^2 T\tilde{\lambda}^* \sqrt{1+s}$
   (ii) $\|\tilde{\mathbf{w}}_S - \mathbf{w}_S^*\|_2 \le C_0^2 (\lambda + T\tilde{\lambda}^*)\sqrt{1+s}$

(b) **Equivalence of Solutions:** *Suppose in addition that the minimum signal strength condition in Theorem 4.3 holds. Then the set of global minimizers for the oracle program and the restricted penalized program are identical.*

*Proof.* The properties hold on the high-probability event $\mathcal{T}_{\tilde{\lambda}^*, T}$.

**Part (a): Estimation Error Bounds**

**(i) Oracle Estimator Bound:** By definition of $\hat{\mathbf{w}}_S^{\mathcal{O}}$, we have the basic inequality for its excess risk $\mathcal{E}(\hat{\mathbf{w}}^{\mathcal{O}}) \le (\mathbb{E}_n - \mathbb{E})(\ell_{\mathbf{w}^*} - \ell_{\hat{\mathbf{w}}^{\mathcal{O}}})$. On the event $\mathcal{T}_{\tilde{\lambda}^*, T}$, this is bounded by:

$$\mathcal{E}(\hat{\mathbf{w}}^{\mathcal{O}}) \le T\tilde{\lambda}^* \left( \|\hat{\mathbf{w}}_{upper}^{\mathcal{O}} - \mathbf{w}_{upper}^*\|_2 + \sum_{j \in S} \|\hat{\mathbf{W}}_{0,j}^{\mathcal{O}} - \mathbf{W}_{0,j}^*\|_2 \right)$$

We apply the Cauchy-Schwarz inequality to the sum of norms. Letting the vector of norms be $v = (\|\cdot\|_2, \ldots, \|\cdot\|_2)$ and a vector of ones be $\mathbf{1}$, we have $v \cdot \mathbf{1} \le \|v\|_2 \|\mathbf{1}\|_2$. This gives:

$$\|\hat{\mathbf{w}}_{upper}^{\mathcal{O}} - \mathbf{w}_{upper}^*\|_2 + \sum_{j \in S} \|\hat{\mathbf{W}}_{0,j}^{\mathcal{O}} - \mathbf{W}_{0,j}^*\|_2 \le \sqrt{1+s} \sqrt{\|\hat{\mathbf{w}}_{upper}^{\mathcal{O}} - \mathbf{w}_{upper}^*\|_2^2 + \sum_{j \in S} \|\hat{\mathbf{W}}_{0,j}^{\mathcal{O}} - \mathbf{W}_{0,j}^*\|_2^2}$$

The term under the square root is precisely $\|\hat{\mathbf{w}}_S^{\mathcal{O}} - \mathbf{w}_S^*\|_2$. Letting $D(\hat{\mathbf{w}}^{\mathcal{O}}) = \|\hat{\mathbf{w}}_S^{\mathcal{O}} - \mathbf{w}_S^*\|_2$, we have $\mathcal{E}(\hat{\mathbf{w}}^{\mathcal{O}}) \le T\tilde{\lambda}^* \sqrt{1+s} D(\hat{\mathbf{w}}^{\mathcal{O}})$. Applying Young's inequality and the quadratic lower bound $\mathcal{E}(\hat{\mathbf{w}}^{\mathcal{O}}) \ge D(\hat{\mathbf{w}}^{\mathcal{O}})^2 / C_0^2$ yields the stated bound.

**(ii) Restricted Estimator Bound:** The restricted penalized program is an instance of the general framework in (1) with a reduced feature set. Therefore, the conclusion of Theorem 1.2(i) applies directly, yielding the stated bound.

**Part (b): Equivalence of Solutions**

We prove that the solution sets are identical by showing that the solutions must have equal loss and penalty values on the high-probability event.

**Step 1: Establish Equality of the Penalty Term**

We show that for any solutions $\hat{\mathbf{w}}_S^{\mathcal{O}}$ and $\tilde{\mathbf{w}}_S$, their total penalty values are identical. This requires proving that all their active weight norms exceed the threshold $\delta\lambda$.

*For the restricted estimator $\tilde{\mathbf{w}}_S$:* By the minimum signal strength condition in Theorem 4.3, for each $j \in S$,

$$\|\mathbf{W}_{0,j}^*\|_2 > \delta\lambda + C_0^2(\lambda + T\tilde{\lambda}^*)\sqrt{1+s}.$$

From Part (a)(ii), the restricted estimator satisfies

$$\|\tilde{\mathbf{W}}_S - \mathbf{w}_S^*\|_2 \leq C_0^2(\lambda + T\tilde{\lambda}^*)\sqrt{1+s}.$$

Since the error of a single group is bounded by the total error,

$$\|\tilde{\mathbf{W}}_{0,j} - \mathbf{W}_{0,j}^*\|_2 \leq C_0^2(\lambda + T\tilde{\lambda}^*)\sqrt{1+s}.$$

Substituting into the signal condition gives

$$\|\mathbf{W}_{0,j}^*\|_2 > \delta\lambda + \|\tilde{\mathbf{W}}_{0,j} - \mathbf{W}_{0,j}^*\|_2.$$

By the reverse triangle inequality,

$$\|\tilde{\mathbf{W}}_{0,j}\|_2 \geq \|\mathbf{W}_{0,j}^*\|_2 - \|\tilde{\mathbf{W}}_{0,j} - \mathbf{W}_{0,j}^*\|_2 > \delta\lambda.$$

*For the oracle estimator $\hat{\mathbf{w}}_S^{\mathcal{O}}$:* From Part (a)(i), the oracle estimator satisfies

$$\|\hat{\mathbf{w}}_S^{\mathcal{O}} - \mathbf{w}_S^*\|_2 \leq C_0^2 T\tilde{\lambda}^*\sqrt{1+s},$$

which is strictly smaller than the bound for $\tilde{\mathbf{w}}_S$ since $\lambda > 0$. Hence, for each $j \in S$,

$$\|\mathbf{W}_{0,j}^*\|_2 > \delta\lambda + C_0^2(\lambda + T\tilde{\lambda}^*)\sqrt{1+s} > \delta\lambda + \|\hat{\mathbf{W}}_{0,j}^{\mathcal{O}} - \mathbf{W}_{0,j}^*\|_2.$$

Applying the reverse triangle inequality again yields

$$\|\hat{\mathbf{W}}_{0,j}^{\mathcal{O}}\|_2 > \delta\lambda.$$

Since all active component norms for both estimators exceed $\delta\lambda$, they lie in the region where the penalty function $\rho_\lambda(t)$ is constant. This proves the equality of their total penalty values:

$$\sum_{j \in S} \rho_\lambda(\|\hat{\mathbf{W}}_{0,j}^{\mathcal{O}}\|_2) = \sum_{j \in S} \rho_\lambda(\|\tilde{\mathbf{W}}_{0,j}\|_2).$$

**Step 2: Establish Equality of the Loss Term and Conclude Equivalence**

By the definition of $\tilde{\mathbf{w}}_S$ as a global minimizer of the penalized objective, its value must be less than or equal to that of any other point, including $\hat{\mathbf{w}}_S^{\mathcal{O}}$:

$$\mathcal{L}_n(\tilde{\mathbf{w}}_S) + \sum_{j \in S} \rho_\lambda(\|\tilde{\mathbf{W}}_{0,j}\|_2) \leq \mathcal{L}_n(\hat{\mathbf{w}}_S^{\mathcal{O}}) + \sum_{j \in S} \rho_\lambda(\|\hat{\mathbf{W}}_{0,j}^{\mathcal{O}}\|_2)$$

Since we have just shown the penalty terms are equal, leaving:

$$\mathcal{L}_n(\tilde{\mathbf{w}}_S) \leq \mathcal{L}_n(\hat{\mathbf{w}}_S^{\mathcal{O}})$$

Furthermore, by the definition of the oracle estimator $\hat{\mathbf{w}}_S^{\mathcal{O}}$ as a global minimizer of the loss, we know that $\mathcal{L}_n(\hat{\mathbf{w}}_S^{\mathcal{O}}) \leq \mathcal{L}_n(\tilde{\mathbf{w}}_S)$. The only way both inequalities can hold is if the loss values are equal:

$$\mathcal{L}_n(\tilde{\mathbf{w}}_S) = \mathcal{L}_n(\hat{\mathbf{w}}_S^{\mathcal{O}})$$

We have now shown that any restricted solution $\tilde{\mathbf{w}}_S$ achieves the global minimum value of the oracle loss. Therefore, any restricted solution is also an oracle solution. Since their loss and penalty values are identical, their penalized objective values are also identical. As $\tilde{\mathbf{w}}_S$ is a global minimizer of the penalized program, $\hat{\mathbf{w}}_S^{\mathcal{O}}$ must also be one. Therefore, any oracle solution is also a restricted solution.

Since both directions of inclusion hold, the sets of global minimizers for the two programs are identical. $\square$

Next we establish two crucial lemmas about the properties of the gradient components.

**Lemma B.4** (Boundedness and Lipschitz continuity of the gradient). *Under Condition 4.3 (bounded third derivative) and with the compact parameter space $\Theta = \{\mathbf{w} \in \mathbb{R}^p : \|\mathbf{w}\|_2 \leq K\}$, the following hold for any fixed $(x, y)$ and any indices $j, k$:*

(a) ***Boundedness:*** *The function $g_{\mathbf{w},j,k}(x,y) = \left[\nabla_{\mathbf{W}_{0,j}} \ell(f_{\mathbf{w}}(x), y)\right]_k$ is bounded on $\Theta$. That is, there exists $B > 0$ such that*
$$|g_{\mathbf{w},j,k}(x,y)| \leq B \quad \text{for all } \mathbf{w} \in \Theta.$$

(b) ***Lipschitz continuity:*** *The map $\mathbf{w} \mapsto g_{\mathbf{w},j,k}(x,y)$ is Lipschitz on $\Theta$. There exists $L_g > 0$ such that for all $\mathbf{w}, \mathbf{w}' \in \Theta$,*
$$|g_{\mathbf{w},j,k}(x,y) - g_{\mathbf{w}',j,k}(x,y)| \leq L_g \|\mathbf{w} - \mathbf{w}'\|_2.$$

*Proof.* **Preliminaries.** Let
$$G(\mathbf{w}) := \nabla_{\mathbf{w}} \ell(f_{\mathbf{w}}(x), y) \in \mathbb{R}^p \qquad \text{and} \qquad H(\mathbf{w}) := \nabla_{\mathbf{w}}^2 \ell(f_{\mathbf{w}}(x), y) \in \mathbb{R}^{p \times p}.$$

Condition 4.3 states that the third derivatives of $\ell \circ f_{\mathbf{w}}$ with respect to $\mathbf{w}$ are uniformly bounded on $\Theta$. In particular, the Hessian $H(\mathbf{w})$ is continuously differentiable on $\Theta$ and hence continuous. Moreover, bounded third derivatives imply that $H(\cdot)$ is Lipschitz on $\Theta$; i.e. there exists $M < \infty$ such that for all $\mathbf{w}, \boldsymbol{\nu} \in \Theta$,

$$\|H(\mathbf{w}) - H(\boldsymbol{\nu})\|_{\text{op}} \leq M \|\mathbf{w} - \boldsymbol{\nu}\|_2,$$

where $\| \cdot \|_{\text{op}}$ denotes the operator norm. The compactness of $\Theta$ then ensures $\sup_{\boldsymbol{\nu} \in \Theta} \|H(\boldsymbol{\nu})\|_{\text{op}} < \infty$.

**Part (a) — Boundedness.** The coordinate function $g_{\mathbf{w},j,k}(x,y)$ is a component of the vector-valued function $G(\mathbf{w})$. From the preliminaries, $G(\mathbf{w})$ is continuous on $\Theta$. Since $\Theta$ is compact, the Extreme Value Theorem implies that each coordinate of $G$ is bounded on $\Theta$. Therefore there exists $B > 0$ such that $|g_{\mathbf{w},j,k}(x,y)| \leq B$ for all $\mathbf{w} \in \Theta$, proving part (a).

**Part (b) — Lipschitz continuity.** Fix $\mathbf{w}, \mathbf{w}' \in \Theta$. Using the fundamental theorem of calculus (integral form) for vector-valued functions,

$$G(\mathbf{w}) - G(\mathbf{w}') \;=\; \int_0^1 H\big(\mathbf{w}' + t(\mathbf{w} - \mathbf{w}')\big) (\mathbf{w} - \mathbf{w}') \, dt.$$

Taking the $(j,k)$-coordinate and absolute values gives

$$|g_{\mathbf{w},j,k}(x,y) - g_{\mathbf{w}',j,k}(x,y)| \;\leq\; \left\| \int_0^1 H\big(\mathbf{w}' + t(\mathbf{w} - \mathbf{w}')\big) \, dt \right\|_{\text{op}} \|\mathbf{w} - \mathbf{w}'\|_2.$$

Because $H(\cdot)$ is continuous on the compact set $\Theta$, its operator norm attains a finite supremum on $\Theta$. Define

$$L_g \;:=\; \sup_{\boldsymbol{\nu} \in \Theta} \|H(\boldsymbol{\nu})\|_{\text{op}} \;<\; \infty.$$

Then

$$\left\| \int_0^1 H\big(\mathbf{w}' + t(\mathbf{w} - \mathbf{w}')\big) \, dt \right\|_{\text{op}} \leq \int_0^1 \|H(\mathbf{w}' + t(\mathbf{w} - \mathbf{w}'))\|_{\text{op}} \, dt \leq L_g,$$

and hence

$$|g_{\mathbf{w},j,k}(x,y) - g_{\mathbf{w}',j,k}(x,y)| \leq L_g \|\mathbf{w} - \mathbf{w}'\|_2.$$

This proves that $g_{\mathbf{w},j,k}(x,y)$ is Lipschitz on $\Theta$ with constant $L_g$, completing the proof of part (b) and of the lemma. $\qquad \square$

**Lemma B.5** (Gradient concentration over a fixed support). *Let $\{(X_i, Y_i)\}_{i=1}^n$ be i.i.d. observations and the function class $\mathcal{F}_S$ be the set of scalar components of the gradients, restricted to the oracle parameter subspace $\Theta_S = \{\mathbf{w} \in \Theta : supp(\mathbf{w}) \subseteq S_{\text{param}}\}$:*

$$\mathcal{F}_S = \big\{ g_{\mathbf{w},j,k}(x,y) := [\nabla_{\mathbf{W}_{0,j}} \ell(f_{\mathbf{w}}(x), y)]_k \ : \ \mathbf{w} \in \Theta_S, \ j = 1, \ldots, d, \ k = 1, \ldots, d_1 \big\}.$$

*Assume there exist $B, L > 0$ and a compact set $\Theta \subset \mathbb{R}^p$ in a ball of radius $K$ such that:*

1. *(Uniform boundedness) for all $g \in \mathcal{F}_S$, $|g(x,y)| \leq B$ a.s.*

2. *(Uniform Lipschitz) for all $j, k, (x,y)$, $|[\nabla_{\mathbf{W}_{0,j}} \ell(f_{\mathbf{w}}(x), y)]_k - [\nabla_{\mathbf{W}_{0,j}} \ell(f_{\mathbf{w}'}(x), y)]_k| \leq L \|\mathbf{w} - \mathbf{w}'\|_2$.*

3. *(Uniform variance bound) $\sup_{g \in \mathcal{F}_S} \text{Var}[g(X, Y)] \leq \sigma^2$.*

*Then there exist absolute constants $C_1, C_2, C_3 > 0$ such that for*

$$\lambda_{GC} := \sqrt{\frac{d^* + \log(d \cdot d_1)}{n}},$$

*the following holds:*

$$\mathbb{P}\Big( \sup_{g \in \mathcal{F}_S} |(\mathbb{P}_n - \mathbb{E})g| \leq C_1 \, B \, \lambda_{GC} \Big) \ \geq \ 1 - C_2 \exp\big( - C_3 \, n\lambda_{GC}^2 \big).$$

*Proof.* The proof proceeds in three steps.

**Step 1 (Covering).** We cover the set of functions $\mathcal{F}_S$. These functions are generated by parameter vectors $\mathbf{w}$ from the set $\Theta_S$. While the vectors in $\Theta_S$ are elements of $\mathbb{R}^p$, they are constrained to a fixed $d^*$-dimensional subspace defined by the support $S_{\text{param}}$. The geometric complexity of this set is therefore equivalent to that of a $d^*$-dimensional ball of radius $K$. The covering number for this set is given by:

$$N(\varepsilon, \Theta_S, \|\cdot\|_2) \leq \left(\frac{3K}{\varepsilon}\right)^{d^*}.$$

By the Lipschitz mapping property, the metric entropy $H(\varepsilon, \mathcal{F}_S, L_2(P))$ is bounded by:

$$H(\varepsilon, \mathcal{F}_S, L_2(P)) \leq \log(d \cdot d_1) + \log N(\varepsilon/L, \Theta_S, \|\cdot\|_2)$$

$$\leq \log(d \cdot d_1) + d^* \log\left(\frac{3KL}{\varepsilon}\right).$$

**Step 2 (Symmetrization and Dudley's Bound).** We use Dudley's entropy integral to bound the Rademacher complexity:

$$\mathbb{E}\mathcal{R}_n(\mathcal{F}_S) \leq \frac{C}{\sqrt{n}} \int_0^B \sqrt{H(u, \mathcal{F}_S, L_2(P))} \, du.$$

Using our entropy bound, the integral is bounded by:

$$\int_0^B \sqrt{\log(d \cdot d_1) + d^* \log(3KL/u)} \, du \leq C' B \sqrt{d^* + \log(d \cdot d_1)}.$$

This yields the expectation bound:

$$\mathbb{E}\Big[ \sup_{g \in \mathcal{F}_S} |(\mathbb{P}_n - \mathbb{E})g| \Big] \lesssim \frac{B}{\sqrt{n}} \sqrt{d^* + \log(d \cdot d_1)} = B\lambda_{GC}.$$

**Step 3 (Concentration to High Probability).** We use Bousquet's inequality, which states that for any $t > 0$, with probability at least $1 - e^{-t}$,

$$\sup_{g \in \mathcal{F}_S} |(\mathbb{P}_n - \mathbb{E})g| \leq E_n + \sqrt{\frac{2\sigma^2 t}{n}} + \frac{2Bt}{3n},$$

where $E_n$ is the expectation bound from Step 2. We choose $t = cn\lambda_{GC}^2$ for a suitable constant $c > 0$. The terms on the right-hand side become:

$$E_n \lesssim B\lambda_{GC}, \quad \sqrt{\frac{2\sigma^2 t}{n}} \lesssim \sigma\sqrt{c}\,\lambda_{GC}, \quad \frac{2Bt}{3n} \lesssim Bc\lambda_{GC}^2.$$

For $n$ large enough, the $\lambda_{GC}^2$ term is dominated by the $\lambda_{GC}$ terms. Thus, there exist constants $C_1, C_2, C_3 > 0$ such that:

$$\mathbb{P}\Big(\sup_{g \in \mathcal{F}_S} |(\mathbb{P}_n - \mathbb{E})g| \leq C_1\,B\,\lambda_{GC}\Big) \geq 1 - C_2 \exp(-C_3 n\lambda_{GC}^2),$$

which completes the proof. $\square$

## B.5 Main Proof

*Proof of Theorem 4.3.* We verify that the augmented oracle estimator $\hat{\mathbf{w}}^{\mathcal{O}} = (\hat{\mathbf{w}}_S^{\mathcal{O}}, \mathbf{0})$ is a stationary point of the penalized program equation 4 by checking the Karush-Kuhn-Tucker (KKT) conditions with high probability.

(a) **Primal feasibility:** $\hat{\mathbf{w}}^{\mathcal{O}} \in \Theta$ is satisfied by construction.

(b) **Stationarity on the active set ($S$):** First, we establish that $\hat{\mathbf{w}}_S^{\mathcal{O}}$ is an interior point of its program. Assuming the true parameters are not on the boundary of the feasible set (i.e., $\|\mathbf{w}_S^*\|_2 \leq K/2$), the interior point property follows from the triangle inequality and the estimation error bound in Lemma B.3(a)(i):

$$\|\hat{\mathbf{w}}_S^{\mathcal{O}}\|_2 \leq \|\mathbf{w}_S^*\|_2 + \|\hat{\mathbf{w}}_S^{\mathcal{O}} - \mathbf{w}_S^*\|_2 \leq \frac{K}{2} + T\tilde{\lambda}^*\sqrt{1+s}C_0^2.$$

For a sufficiently large sample size $n$, the second term is bounded by $K/2$, ensuring that $\|\hat{\mathbf{w}}_S^{\mathcal{O}}\|_2 < K$. As an interior-point solution, its unpenalized loss gradient must be zero: $\left[\nabla\mathcal{L}_n(\hat{\mathbf{w}}^{\mathcal{O}})\right]_S = \mathbf{0}$.

Furthermore, Lemma B.3(b) establishes the equivalence of the oracle and the restricted penalized programs. This means $\hat{\mathbf{w}}_S^{\mathcal{O}}$ is also a global minimizer of the restricted penalized program and must satisfy its KKT conditions. The stationarity condition for any $j \in S$ is $\nabla_{\mathbf{W}_{0,j}}\mathcal{L}_n(\hat{\mathbf{w}}^{\mathcal{O}}) + \mathbf{z}_j = \mathbf{0}$, where $\mathbf{z}_j$ is a vector in the subgradient of the penalty. Since the loss gradient is zero, the penalty subgradient $\mathbf{z}_j$ must also be zero. Therefore, the stationarity condition for the full penalized program equation 4 is satisfied for all weights in the active set $S$.

(c) **Strict dual feasibility on the inactive set ($S^c$):** For any $j \in S^c$, we must show that $\|\nabla_{\mathbf{W}_{0,j}}\mathcal{L}_n(\hat{\mathbf{w}}^{\mathcal{O}})\|_2 < \lambda$. We decompose the gradient into its stochastic and deterministic parts:

$$\nabla\mathcal{L}_n(\hat{\mathbf{w}}^{\mathcal{O}}) = \underbrace{\left(\nabla\mathcal{L}_n(\hat{\mathbf{w}}^{\mathcal{O}}) - \nabla\mathcal{L}(\hat{\mathbf{w}}^{\mathcal{O}})\right)}_{\text{(I) Stochastic Error}} + \underbrace{\left(\nabla\mathcal{L}(\hat{\mathbf{w}}^{\mathcal{O}}) - \nabla\mathcal{L}(\mathbf{w}^*)\right)}_{\text{(II) Bias}}.$$

**Bounding (I): Stochastic Error.** The conditions required to apply Lemma B.5—uniform boundedness, uniform Lipschitz continuity, and a uniform variance bound—follow directly from Lemma B.4. Consequently, Lemma B.5 applies and guarantees that, with high probability, the stochastic error is uniformly controlled:

$$\max_{j \in S^c} \|\nabla_{\mathbf{W}_{0,j}}(\mathcal{L}_n - \mathcal{L})(\hat{\mathbf{w}}^{\mathcal{O}})\|_2 \leq \sqrt{d_1}\sup_{g \in \mathcal{F}_S}|(\mathbb{P}_n - \mathbb{E})g| \leq \sqrt{d_1}C_4\lambda_{GC},$$

for some constant $C_4$, where the statistical rate $\lambda_{GC}$ is of order $O(\sqrt{\log(d)/n})$.

**Bounding (II): Bias.** The population gradient is Lipschitz continuous (Lemma B.4), so $\|\nabla\mathcal{L}(\hat{\mathbf{w}}^{\mathcal{O}}) - \nabla\mathcal{L}(\mathbf{w}^*)\|_2 \leq L\|\hat{\mathbf{w}}^{\mathcal{O}} - \mathbf{w}^*\|_2$. Since both $\hat{\mathbf{w}}^{\mathcal{O}}$ and $\mathbf{w}^*$ are zero on the inactive set $S^c$, this distance simplifies to $\|\hat{\mathbf{w}}_S^{\mathcal{O}} - \mathbf{w}_S^*\|_2$. Using the sharp estimation error bound for the oracle estimator from Lemma B.3(a)(i), we can bound the bias:

$$\|\nabla\mathcal{L}(\hat{\mathbf{w}}^{\mathcal{O}}) - \nabla\mathcal{L}(\mathbf{w}^*)\|_2 \leq L \cdot C_0^2 T\tilde{\lambda}^*\sqrt{1+s}.$$

Crucially, the rate of the restricted complexity term $\tilde{\lambda}^*$ depends on the oracle dimension $d^*$ (which scales with $s$), not the ambient feature dimension $d$.

**Combining the Bounds and Concluding the Proof.** By combining these results, the total gradient norm for any inactive feature $j \in S^c$ is bounded by the sum of the two parts:

$$\|\nabla_{\mathbf{W}_{0,j}} \mathcal{L}_n(\hat{\mathbf{w}}^{\mathcal{O}})\|_2 \leq \underbrace{\sqrt{d_1} C_4 \lambda_{GC}}_{\text{Stochastic Error}} + \underbrace{LC_0^2 T \tilde{\lambda}^* \sqrt{1+s}}_{\text{Bias}}.$$

In the high-dimensional setting where $d \gg s$, the stochastic error term dominates because its rate $\lambda_{GC}$ scales with $\sqrt{\log d}$, while the bias term's rate $\tilde{\lambda}^*$ scales with the much smaller oracle dimension $\sqrt{\log d^*}$. The entire gradient norm is therefore bounded by an expression of order $O_p(\sqrt{\log(d)/n})$. As established in Lemma B.5, the gradient concentration holds on an event with probability at least $1 - O(\exp(-C \log d))$ for some constant $C > 0$. Thus, by selecting a tuning parameter $\lambda$ that satisfies $\lambda \geq C_5 \sqrt{\log(d)/n}$ for a large enough constant $C_5$, we ensure that with high probability, the gradient norm is smaller than the penalty threshold:

$$\|\nabla_{\mathbf{W}_{0,j}} \mathcal{L}_n(\hat{\mathbf{w}}^{\mathcal{O}})\|_2 < \lambda.$$

This establishes strict dual feasibility for all $j \in S^c$, completing the proof.

Since all KKT conditions are satisfied with high probability, the augmented oracle estimator $\hat{\mathbf{w}}^{\mathcal{O}}$ is a stationary point of the penalized program. $\square$

## C  Complete Results for Survival Model

Figure 7 shows that larger variations in C-index are associated with larger censoring rates overall. GMCPNet and GSCADNet achieve comparable results to Oracle-NN while surpassing all other methods, including Oracle-RSF.

## D  Simulation with Correlated Variables

The simulation study in Section 5 focuses on independent covariates. However, in real-world applications, particularly in high-dimensional settings, the presence of correlations among covariates is common and presents a challenge for feature selection. In this section, we assess the effectiveness of the proposed method using simulated data that incorporates correlated variables.

To be more specific, we extend the high-dimensional scenario described in Section 5 by generating a correlated covariate vector, denoted as $\mathbf{X} \sim N(0, \Sigma)$. The correlation structure is defined using a power decay pattern, where $\Sigma_{ij} = 0.5^{|i-j|}$. This modification allows us to examine the performance of our method in the presence of correlation among the covariates. Comparing the results of feature selection for independent covariates in Table 1 to the outcomes presented in Table 2, it becomes evident that STG and GLSSONet exhibit larger variations in selected model sizes, along with FNR and FPR in the regression model. This behavior can be attributed to the presence of correlated features. In contrast, the proposed GMCPNet and GSCADNet methods effectively identify relevant variables while maintaining relatively low false positive and negative rates across all models. Furthermore, Figure 8 demonstrates that both GMCPNet and GSCADNet perform comparably to the Oracle-NN method in the regression and survival models, while outperforming other non-oracle approaches in the classification model. These findings indicate that the proposed methods exhibit robustness against correlations among covariates in terms of feature selection and model prediction.

## E  Understanding the Impact of LASSO Bias on Feature Selection

Our numerical study demonstrates that GLASSONet tends to select many noisy variables due to the bias introduced by the LASSO penalty. In contrast, the group-concave regularization used in our proposed framework (e.g., GMCPNet and CSCADNet) reduces this bias and improves feature selection accuracy.

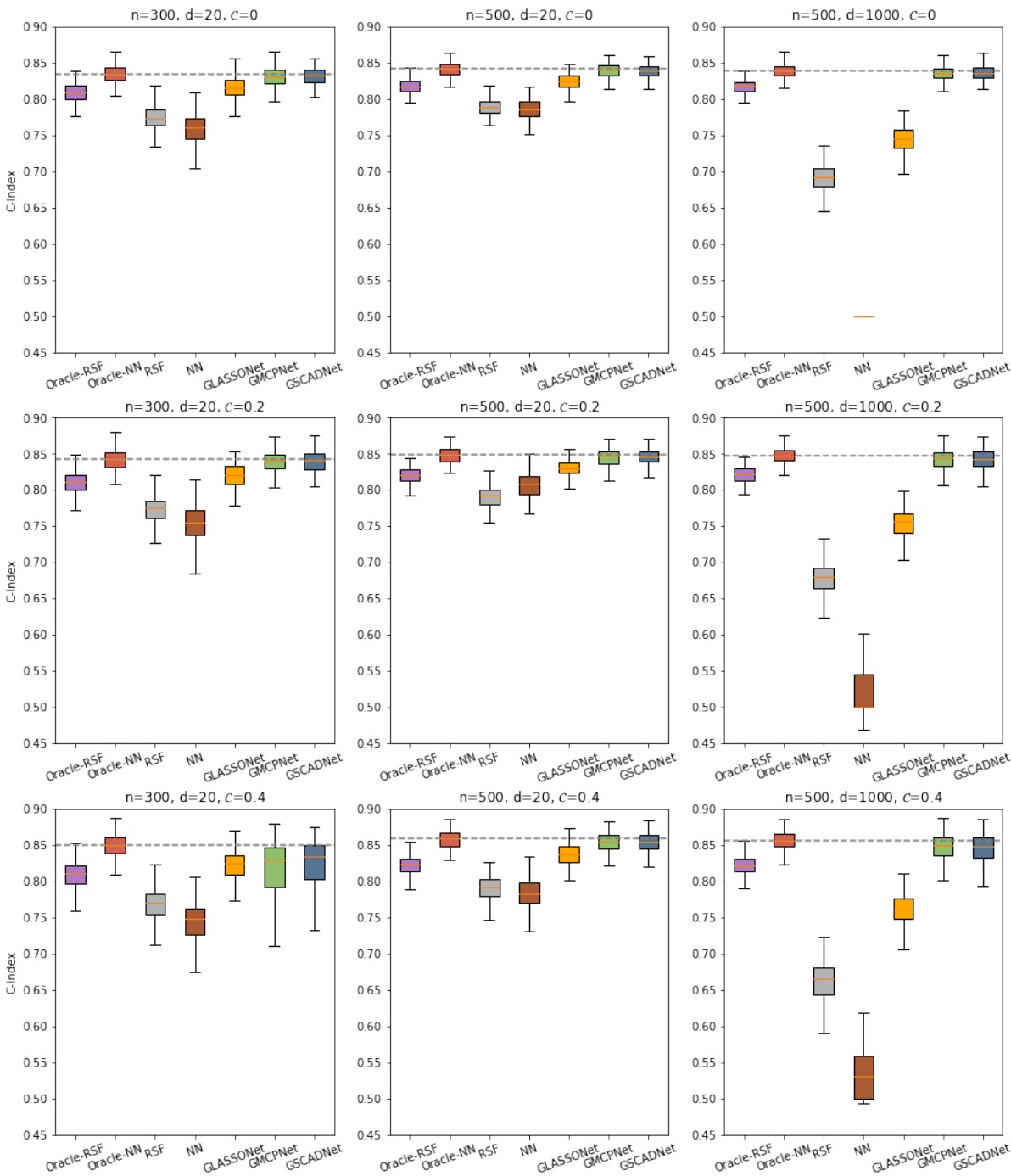

Figure 7: **C-Index of the proposed methods for the survival model outlined in Example 5.3.**
The dashed line represents the median C-Index of the Oracle-NN, used as a benchmark for comparison.

Table 2: **Feature selection results of STG, GLASSONet, GMCPNet, and GSCADNet using correlated features in high-dimensional scenario** ($n = 500, d = 1000$)**.** The False positive rate (FPR %), False negative rate (FNR %), and model size (MS) with standard deviation (SD) in parentheses are displayed.

| Method | Regression | | Classification | | Survival ($\mathcal{C} = 0.2$) | |
|---|---|---|---|---|---|---|
| | FPR, FNR | MS (SD) | FPR, FNR | MS (SD) | FPR, FNR | MS (SD) |
| STG | 8.4, 16.6 | 86.8(132.6) | 1.5, 21.0 | 18.6(121.1) | -, - | -(-) |
| GLASSONet | 28.8, 26.6 | 290.0(144.6) | 19.3, 22.4 | 195.7(116.4) | 16.0, 1.9 | 163.1(51.4) |
| GMCPNet | 0.1, 13.4 | 4.0(1.4) | 0.2, 13.9 | 5.5(4.5) | 0.0, 0.0 | 4.1(0.4) |
| GSCADNet | 0.1, 13.2 | 4.0(1.2) | 0.1, 11.8 | 4.8(2.9) | 0.0, 0.0 | 4.1(0.6) |

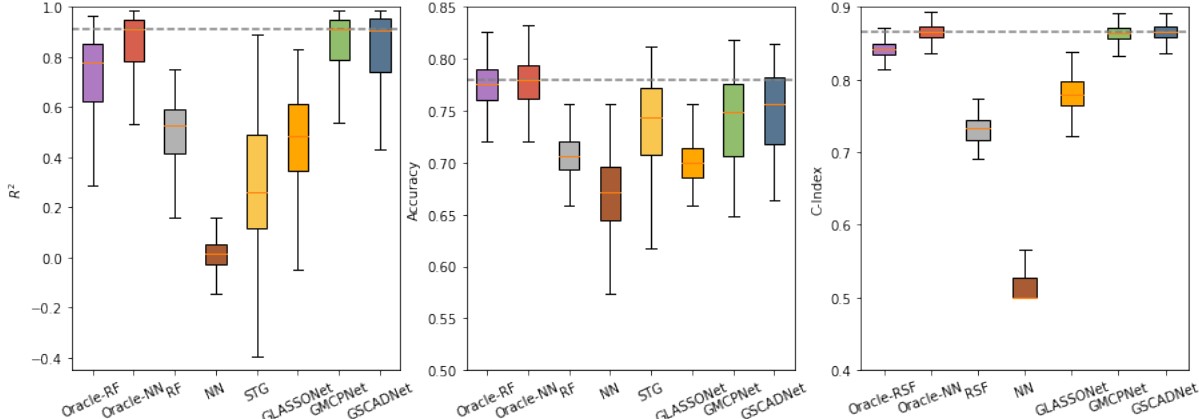

Figure 8: **Prediction scores of the proposed methods for the regression, classification, and survival models** ($\mathcal{C} = 0.2$) **using correlated features in high-dimensional scenario** ($n = 500, d = 1000$)**.** The dashed lines represent the median score of the Oracle-NN, used as a benchmark for comparison.

To further investigate the impact of LASSO's bias on feature selection, we propose a modified version of GLASSONet, applying a relaxed LASSO approach and terming it relaxed-GLASSONet.

The relaxed-GLASSONet method follows a two-stage procedure: for each group LASSO parameter $\lambda$, we first select features using GLASSONet. Then, we refit a standard neural network with only the selected features by setting $\lambda = 0$, thereby reducing bias during model fitting. The final model is selected based on its predictive performance on a validation set. Our goal is to explore whether the relaxed-GLASSONet can mitigate the feature overselection observed in the LASSO-regularized approach by removing the bias, and ultimately enhance prediction performance.

We apply the relaxed-GLASSONet method to synthetic data generated from the XOR-type signal regression model, repeating the simulation described in Section 5.1. We compare relaxed-GLASSONet with GLAS-SONet, as well as our proposed GMCPNet and GSCADNet methods, with results presented in Figure 9. Our results indicate that relaxed-GLASSONet selects significantly fewer false positives across varying feature counts, thereby improving prediction accuracy. Importantly, relaxed-GLASSONet performs comparably to GMCPNet and GLASSONet in low-dimensional settings, but its performance declines as dimensionality increases. These findings confirm that the LASSO penalty tends to over-select features to compensate for its inherent bias. This overselection can be mitigated by reducing bias through model refitting with only the selected features, leading to more accurate feature selection.

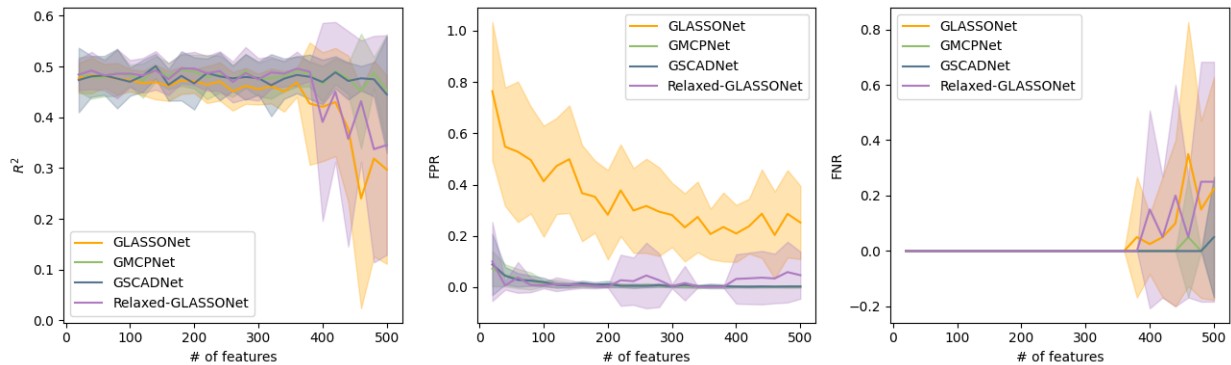

Figure 9: **Simulation results of relaxed-GLASSONet for XOR-Type signal model.** The $R^2$ scores, false positive rate (FPR), and false negative rate (FNR) are presented in the left, middle, and right panels, respectively. The central lines are the means while the shaded areas represent standard deviations.

## F  Implementation Details

### F.1  Simulation studies

To ensure a fair comparison among all the neural-net-based methods, we adopted a ReLU-activated Multi-Layer Perceptron (MLP) with two hidden layers consisting of 10 and 5 units, respectively. The network weights were initialized by sampling from a Gaussian distribution with mean 0 and standard deviation 0.1, while the bias terms were set to 0 following the Xavier initialization technique (Glorot & Bengio, 2010). The optimization of the neural networks was performed using the Adam optimizer with a base learning rate (LR) of 0.001.

For all the methods falling within the framework of Equation (1) in the paper, we selected the optimal values of $\lambda$ and $\alpha$ from a two-dimensional grid, with $\lambda$ and $\alpha$ ranging over 50 and 10 evenly spaced values on a logarithmic scale, respectively. The selection was based on their performance on the validation set, which consisted of 20% of the training set. The parameter search ranges are displayed in Table 3. We set $\lambda = 0$ for NN and Oracle-NN to deactivate feature selection. For GLASSONet, GMCPNet, and GSCADNet, the number of epochs at $\lambda_{min}$ was set to 2000 for the LD and 200 for the HD scenarios. For all other values of

$\lambda$, the number of epochs was set to 200 for both LD and HD settings. The number of epochs for NN was consistently fixed at 5000.

We employed RF with 1000 decision trees for the model fitting process. We implemented the STG method as described in Yamada et al. (2020) that the LR and regularization parameter $\lambda$ were optimized via Optuna with 500 trials, using 10% of the training set as a validation set. The number of epochs was 2000 for each trial.

Table 3: **List of the search range for the tuning parameters used in our simulation.**

| Param | Search range | |
|---|---|---|
| | LD | HD |
| $\lambda$ | [1e-3, 0.5] | [1e-2, 0.5] |
| $\alpha$ | [1e-3, 0.1] | [1e-2, 0.1] |
| LR (STG) | [1e-4,0.1] | [1e-4, 0.1] |
| $\lambda$ (STG) | [1e-3, 10] | [1e-2, 100] |
| $\lambda$ (LASSONet) | [5e-4, 2e-3] | [5e-4, 2e-3] |

### F.2 Real Data Example

In the analysis of real data examples, the implementation details remain the same as the HD scenario in the simulation studies, with the following modifications:

- For the survival analysis on the CALGB-90401 dataset, we utilized the MLP with two hidden layers, each consisting of 10 nodes. In hyperparameter tuning, we explored 100 values of $\lambda$ ranging from 0.01 to 0.1 for GMCPNet and GSCADNet. Additionally, we increase the number of candidates for $\alpha$ to 50.

- In the classification task on the MNIST dataset, we adjust the search range of $\alpha$ to [1e-3, 0.1].

The data from CALGB 90401 is available from the NCTN Data Archive at `https://nctn-data-archive.nci.nih.gov/`. The MNIST dataset was downloaded using the built-in torchvision.datasets.MNIST interface in PyTorch, which automatically retrieves the original dataset from Yann LeCun's repository.

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
