# OpenReview forum: "Sparse-Input Neural Network using Group Concave Regularization"
_TMLR — Accepted by TMLR_

### Review · Reviewer_Vnaj · 2025-07-12

**Summary Of Contributions:**

This paper proposes conducting input variable selection using a combined penalty, which is a combination of an L2-penalty and a group concave penalty. While empirical results are presented, no theoretical results are provided. The overall contribution of the paper appears to be incremental, considering the sparse input work [1]  cited and other existing sparse deep learning methods, see e.g., [2].

[1] Feng, J. and Simon, N. (2017). Sparse input neural networks for high-dimensional nonparametric regression and classification. arXiv:1711.07592.

[2] Sun, Y., Song, Q., & Liang, F. (2021). Consistent Sparse Deep Learning: Theory and Computation, arXiv:2102.13229.

**Audience:**

No

**Claims And Evidence:**

No

**Requested Changes:**

1, In recent years, theoretical guarantees have been developed for sparse deep neural networks, such as in references [2] and [3]. I believe these works merit discussion in the paper.

[2] Sun, Y., Song, X., & Liang, F. (2021). Consistent Sparse Deep Learning: Theory and Computation, arXiv:2102.13229.
[3] Lederer, J. (2022). Statistical guarantees for sparse deep learning, arXiv:2212.05427.

2. The impacts of α and 𝜆 on the performance of the proposed method need to be assessed. For example, if α=0, the method may fail to select any meaningful input variables, given the universal approximation ability of deep neural networks. I understand that the authors determine their values using a validation method. Are there any insights regarding the choice of their values?

3. The comparsions with more baseline methods might be helpful to strengrgen the paper due to the lack of theoretical guarantees.

**Strengths And Weaknesses:**

Strength: The numerical experiments are detailed.

Weakness: No theoretial guarantees are provided for the proposed method.

---

> ### Author Response · Authors · 2025-09-13
> **Authors' Response to Reviewer Comments (part 1)**
>
> We sincerely thank the reviewer for their time and for providing thoughtful feedback that has guided us in strengthening our manuscript. We have addressed all comments, and we believe the paper is significantly improved as a result. Below, we provide a point-by-point response.
>
> On the Summary of Contributions and Weaknesses
>
> Reviewer's Comment: "This paper proposes conducting input variable selection using a combined penalty... While empirical results are presented, no theoretical results are provided. The overall contribution of the paper appears to be incremental... Weakness: No theoretical guarantees are provided for the proposed method."
>
> Our Response: We thank the reviewer for this critical assessment. We agree completely that the original manuscript's main limitation was the lack of theoretical justification. Your feedback motivated us to undertake a rigorous theoretical analysis of our proposed framework, which has now become a central contribution of the paper.
>
> To address this, we have added a comprehensive new section, Section 4: Theoretical Properties. This section now provides the formal guarantees the reviewer correctly identified as missing. Specifically:
>
> 	(1) We establish non-asymptotic, finite-sample guarantees for our estimator's performance (Theorem 4.1), providing explicit bounds on its prediction error (excess risk) and parameter estimation error.
>
> 	(2) We prove that our method possesses the desirable oracle property (Theorem 4.3) under standard high-dimensional conditions. This demonstrates that our estimator can correctly identify the true sparse model and produce nearly unbiased estimates, providing a clear theoretical advantage over the group LASSO penalty, which is known to lack this property.
>
> We believe this major addition elevates the paper's contribution from "incremental" to substantial, as it now provides not only a practical algorithm but also the formal statistical foundation explaining why it performs well.
>
> Action Taken:
> 	(1) A new Section 4: Theoretical Properties (pages 8-14) has been added to the manuscript.
> 	(2) The Abstract and Introduction have been updated to reflect these new theoretical contributions.
>
>
> On Requested Changes
>
> 1. Discussion of Recent Theoretical Work
>
> Reviewer's Comment: "In recent years, theoretical guarantees have been developed for sparse deep neural networks, such as in references [2] and [3]. I believe these works merit discussion in the paper."
>
> Our Response: We thank the reviewer for pointing us to this important and relevant literature. We agree that discussing these works provides valuable context for our own theoretical contributions. We have now incorporated a discussion of Sun et al. (2021) and Lederer (2022) into our paper's introduction.
>
> Action Taken: We have added a paragraph in the Introduction (page 2) that cites and briefly discusses the contributions of Sun et al. (2021) and Lederer (2022), positioning our work within the landscape of recent theoretical advancements in sparse deep learning.
>
>
> 2. Impact and Choice of Hyperparameters α and λ
>
> Reviewer's Comment: "The impacts of α and 𝜆 on the performance of the proposed method need to be assessed... Are there any insights regarding the choice of their values?"
>
> Our Response: We thank the reviewer for this excellent question and for prompting us to clarify the crucial roles of λ and α. We have expanded the manuscript to provide both theoretical and conceptual insights.
>
> 	(1) Regarding λ: This parameter directly controls the model's sparsity by governing the strength of the group concave penalty. As visually demonstrated in the solution path in Figure 1, increasing λ shrinks more groups of weights to zero, leading to sparser models. Our new theoretical analysis in Section 4 provides further insight, suggesting that setting λ on the order of O(\sqrt{log(d)/n}) allows the estimator to achieve the statistical guarantees we establish.
>
> 	(2) Regarding α: The reviewer's intuition is spot on. The ridge penalty, α*\||w| |_2^2  $\alpha \|\bw\|_2^2$, is essential for model stability and effective feature selection. As we have now reinforced in Section 2.2, this term prevents the network from bypassing the input-layer penalty by shrinking input weights while inflating weights in subsequent layers. From a theoretical perspective, this penalty is equivalent to the norm constraint placed on the full parameter vector in our analysis (Section 4.1), which is necessary for deriving our error bounds.
>
> Action Taken: We have reinforced the explanation of the crucial role of α in Section 2.2 to make its importance more explicit.

---

> ### Author Response · Authors · 2025-09-13
> **Authors' Response to Reviewer Comments (part 2)**
>
> 3. Comparison with More Baseline Methods
>
> Reviewer's Comment: "The comparisons with more baseline methods might be helpful to strengthen the paper due to the lack of theoretical guarantees."
>
> Our Response: We appreciate this suggestion. The reviewer rightly pointed out that the original manuscript's primary weakness was its lack of theory. In this revision, we focused our efforts on addressing this fundamental issue by adding the comprehensive theoretical analysis in Section 4. We believe that providing rigorous statistical guarantees for our method is the most significant way to strengthen the paper and address the core of the reviewer's concerns about the paper's contribution.
>
> While adding more baselines is always beneficial, we feel that the current set of comparisons (including a standard Neural Network, Random Forest, the state-of-the-art STG method, and idealized Oracle versions) is diverse and provides a strong benchmark for evaluating our method's performance. Given the major theoretical additions, we hope the reviewer agrees that the paper is now substantially strengthened.
>
> Action Taken: We have focused on adding the new theoretical section as the primary means of strengthening the paper, which directly resolves the reviewer's main concern. The baseline comparisons remain as they were, and we have added a clarification at the beginning of Section 5 explaining the rationale for choosing each of the baseline methods to provide a comprehensive and multifaceted evaluation of our proposed framework.

---

### Review · Reviewer_t1FU · 2025-07-14

**Summary Of Contributions:**

This work provides a new framework for feature selection and prediction in high-dimensional settings, for several tasks and losses (regression, classification, survival analysis). The authors propose and compare several losses for the selection of features, and illustrate their properties in several simulated and real data situations.

**Audience:**

Yes

**Broader Impact Concerns:**

Without strong claims of performance compared to the state of the art, nor theory, the impact of this work is limited.

**Claims And Evidence:**

Yes

**Requested Changes:**

as the authors point out themselves, there is a strong lack of theoretical grounds for the proposed method, and this should be compensated for with a much stronger empirical analysis. Here are two main suggestions
- include baselines, both from the framework that predicts and selects in a single algorithm, but also that does variable selection in a post-hoc way (variable importance, knockoffs, there is a huge litterature). The authors may look into the International Seminar on Selective Inference for references https://sites.google.com/view/selective-inference-seminar/home?authuser=0
- For all methods and experiments, the authors should include more metrics, in particular, proper metrics. Accuracy or the c-index are known to provide a very limited view of the problem (https://arxiv.org/abs/2506.02075 and https://arxiv.org/pdf/2412.10288 are good introductions to this question).

The objective would be to show empirically that the proposed approaches are SOTA both in prediction and selection; otherwise, it is not clear what the point is of using this combined approach compared to existing methods. A study on the computing cost can also be done, which could be a strong point of this method.

**Strengths And Weaknesses:**

Overall, this work seems well motivated and technically sound. It is also clear to read.

---

> ### Author Response · Authors · 2025-09-13
> **Authors' Response to Reviewer Comments (part 1)**
>
> We are grateful to the reviewer for their positive assessment of our work's motivation and clarity, and for their thoughtful suggestions on how to strengthen it. The feedback has been instrumental in guiding our revisions.
>
> The reviewer correctly pointed out that the original manuscript's primary weakness was its lack of theoretical grounds, suggesting that this be compensated for with a much stronger empirical analysis. We have taken this feedback very seriously and have chosen to address the primary concern directly and, we believe, more fundamentally. The most substantial revision to our manuscript is the addition of a comprehensive new theoretical section (Section 4). We believe that providing a rigorous theoretical foundation for why our method works is a more significant improvement than further expanding the empirical study. This new section provides formal, non-asymptotic guarantees for our method's prediction accuracy and estimation error. Crucially, it also provides strong theoretical support for the method's variable selection consistency by showing that the ideal oracle estimator is a stationary point of our objective function. This new theory now provides the strong backing that the reviewer correctly identified as missing.
>
> Below, we address the specific suggestions in detail.
>
> On Requested Changes
>
> 1. Inclusion of More Baselines (e.g., post-hoc methods)
>
> Reviewer's Comment: "include baselines, both from the framework that predicts and selects in a single algorithm, but also that does variable selection in a post-hoc way (variable importance, knockoffs...)"
>
> Our Response: We thank the reviewer for this suggestion. While post-hoc methods like knockoffs are indeed powerful, our work is situated within the literature on embedded methods, which are designed to perform variable selection and function estimation simultaneously within a single, unified framework. Therefore, our chosen baselines (GLASSONet, STG, etc.) represent the most direct and relevant competitors within this specific paradigm. Given the substantial addition of the theoretical section—which provides a formal answer to why an embedded approach with concave penalties is effective—we believe the current set of comparisons provides a clear and sufficient demonstration of our method's performance relative to its direct peers.
>
> Action Taken: We have added a clarification at the beginning of Section 5 explaining the rationale for choosing each of the baseline methods to provide a comprehensive evaluation against the most relevant competitors.
>
> 2. Inclusion of More Performance Metrics
>
> Reviewer's Comment: "For all methods and experiments, the authors should include more metrics, in particular, proper metrics... Accuracy or the c-index are known to provide a very limited view of the problem..."
>
> Our Response: We thank the reviewer for this thoughtful point and for the references. We agree that no single metric provides a complete picture of performance. The metrics used in our study (R-squared, Accuracy, C-index, FPR, and FNR) were chosen because they are standard, widely understood, and the most common metrics used for evaluation in their respective domains. This allows our results to be clearly interpreted and easily compared to a large body of existing work. We believe that the addition of our new theoretical guarantees, which formally establish our method's consistency, provides a more fundamental validation of our approach than the inclusion of additional performance metrics would.
>
> Action Taken: We have focused our revisions on adding the new theoretical section, which provides a formal justification for the method's effectiveness, a more substantial improvement than adding further metrics. The metrics used remain standard for the field to ensure clarity and comparability.

---

> ### Author Response · Authors · 2025-09-13
> **Authors' Response to Reviewer Comments (part 2)**
>
> 3. Demonstrating SOTA Performance and Computing Cost
>
> Reviewer's Comment: "The objective would be to show empirically that the proposed approaches are SOTA both in prediction and selection... A study on the computing cost can also be done..."
>
> Our Response: We agree that demonstrating top-tier performance is crucial. Our simulation results show that our method's performance is often comparable the idealized Oracle-NN benchmark, which is trained with prior knowledge of the true features. This, combined with our new theoretical guarantees (specifically, showing the oracle estimator is a stationary point of our objective), provides strong evidence that our method operates near the theoretical optimum for this task.
>
> Regarding computational cost, we appreciate the reviewer pointing this out as a potential strength. As detailed in the Discussion section (Section 7), in the paragraph beginning, "The runtime of our proposed...",  our analysis shows that the runtime of solving for an entire solution path can be comparable to, or even more efficient than, training a single dense network.
>
> Action Taken:
> (1)	We have reinforced our discussion in the Results section (Section 5.2.1) to explicitly highlight our method's competitive performance by linking the empirical results to the Oracle-NN benchmark and our new theory. Specifically, we have added the following text: This strong empirical evidence, in which our proposed estimators perform on par with the Oracle-NN possessing prior knowledge of the true features, aligns with the oracle properties established in our theoretical analysis (Theorem 4.3).
> (2) 	We have ensured that our analysis of the computational efficiency is clearly presented in the Discussion section (Section 7).
>
> 4. On Broader Impact
>
> Reviewer's Comment: "Without strong claims of performance compared to the state of the art, nor theory, the impact of this work is limited."
>
> Our Response: We thank the reviewer for this comment, which was the primary motivation for our revision. The reviewer noted that without theory or strong performance claims, the impact would be limited. We have now addressed both points. By adding the new theoretical section (Section 4), which provides a rigorous foundation for our method, including formal, non-asymptotic guarantees for its prediction accuracy, estimation error, and oracle property, and by demonstrating performance comparable to an oracle benchmark, we have now provided both the formal guarantees and the strong empirical evidence needed to establish the broad impact and utility of our proposed framework.

---

### Review · Reviewer_tuMf · 2025-08-19

**Summary Of Contributions:**

This article proposes a sparse input feature selection method to improve the performance of neural networks.
In the litterature of feature selection, many methods are developed for linear models
and they may not able to capture complex input-output relations. To model complex relationships,
regularized neural networks are studied, but existing works are limited either to regression-only problems
or to non-sparse feature selection. This article proposes a way to encounter these limitations by using group concave regularization
similar to group LASSO on both synthetic and real datasets.

**Audience:**

Yes

**Claims And Evidence:**

No

**Requested Changes:**

- Fig 1. Please explain what smoother solution paths mean. It is not clear as the x-axis uses a different choice of log (lambda). It would be better to use the same x-axis to compare. Also what is the difference between the curves with the same color in Fig 1?
- According to Section 4, Adam method is used in numerical results as the optimization algorithm. Could you clarify in what sense it is stable and why?
- The comparison with SOTA in Section 5 seems to be of limited impact. In the MNIST case, the test accuracy is not so high compared to convolutional neural networks. At the current test accuracy level, linear SVM could also learn something like Fig 6. Why is this method not compared? What is the SOTA AUC performance in the survival dataset of Section 5.1 ? I think that these SOTA performances should be mentioned as a too large gap suggests that the considered neural networks in the article are not able to capture complex input-output relationships (which could be future work).

**Strengths And Weaknesses:**

Strength :
- The article is well written and easy to follow.
- The way to specify a group for sparse feature selection is a novelty compared to existing works, such as Feng & Simon 2017 on regularized neural networks.
- Empirical validation across diverse data types.

Weakness :
- Although the authors claim that a stable optimization algorithm is proposed in the introduction, it is not so clear what this means. In particular, there is a lack of convergence analysis for the optimization algorithm used in the numerical results.

- The baseline methods chosen in real data example in Section 5 seem to be too weak. It would be better to discuss whether there are other baseline methods to consider.

---

> ### Author Response · Authors · 2025-09-13
> **Authors' Response to Reviewer Comments (part 1)**
>
> We are grateful to the reviewer for their careful reading and for their positive and encouraging feedback on our work. We especially appreciate that they found the article well-written and recognized the novelty of our approach. Their detailed suggestions are extremely helpful, and we have revised the manuscript to address each of them.
>
> On Weaknesses
>
> 1. On the "Stable Optimization Algorithm" and Lack of Convergence Analysis
>
> Reviewer's Comment: "Although the authors claim that a stable optimization algorithm is proposed in the introduction, it is not so clear what this means. In particular, there is a lack of convergence analysis for the optimization algorithm used in the numerical results."
>
> Our Response: We thank the reviewer for pointing out this lack of clarity. We apologize for the confusing wording. Our claim of "stability" does not refer to the convergence guarantees of the Adam optimizer itself for a fixed λ. Proving convergence for optimizers in non-convex deep learning settings is a profound challenge in its own right and beyond the scope of our paper's contribution.
> Instead, our claim of stability refers specifically to the solution path generated by our backward path-wise optimization strategy. As we now clarify in the manuscript, this strategy ensures that small changes in the regularization parameter λ lead to small, predictable changes in the model's weights and sparsity level. This is in contrast to non-pathwise or forward-pathwise approaches, which can produce erratic and unstable solution paths, as illustrated in Figure 1.
>
> Action Taken:
>
> 	(1) We have removed the potentially confusing phrase "stable optimization algorithm" from the introduction's list of contributions on Page 3 and replaced it with a more accurate description: "An effective optimization strategy for generating stable solution paths."
>
> 	(2) We have revised Section 3.2 to explicitly define what we mean by a stable solution path and to clarify that this stability is a product of the backward path-wise strategy. Specifically, we have added the following explanation:
> ‘The goal is to generate a stable solution path, which we define as a sequence of solutions where small changes in the regularization parameter λ result in correspondingly small changes in the model's weights and sparsity level, avoiding the large, erratic jumps often seen with random initializations for each λ (see Figure 1). It is important to clarify that the stability of the path is a product of this overall strategy, not a property of the optimizer (e.g., Adam) used for any single value of $\lambda$.’
>
> 2. On Baselines in Real Data Examples
>
> Reviewer's Comment: "The baseline methods chosen in real data example in Section 5 seem to be too weak. It would be better to discuss whether there are other baseline methods to consider."
>
> Our Response: We thank the reviewer for this point and agree that the choice of baselines warrants clarification. The primary goal of our real-data examples was not to outperform every possible model type, but rather to demonstrate our framework's capabilities in complex, high-dimensional settings and to specifically compare the behavior of different sparse neural network regularization methods (GLASSO vs. GMCP/GSCAD).
>
> For this reason, our main competitor is GLASSONet, with the standard NN serving as a baseline to show the necessity of feature selection. We also included Random Forest because it serves as a powerful and widely-used non-parametric benchmark from a different class of models (tree-based ensembles). Comparing against a strong, non-neural network method like RF demonstrates that our proposed framework is competitive in a broader machine learning context.
>
> Action Taken: We have added a sentence at the beginning of Section 6 (Real Data Example) to clarify that the goal of the comparisons is to differentiate between sparse neural network approaches and to benchmark against strong, standard machine learning methods. Specifically, we have added the following clarification:
>
> ‘In this section, we apply our proposed framework to two real-world datasets to demonstrate its practical utility. The goal of these examples is to compare the behavior of different sparse neural network penalties and benchmark our method against strong, standard machine learning approaches, rather than to achieve absolute state-of-the-art performance on these specific tasks.’

---

> ### Author Response · Authors · 2025-09-13
> **Authors' Response to Reviewer Comments (part 2)**
>
> On Requested Changes
>
> 1. On Figure 1 and Solution Paths
>
> Reviewer's Comment: "Fig 1. Please explain what smoother solution paths mean. It is not clear as the x-axis uses a different choice of log (lambda). It would be better to use the same x-axis to compare. Also what is the difference between the curves with the same color in Fig 1?"
>
> Our Response: Thank you for these detailed questions, which have helped us improve the clarity of Figure 1.
>
> •	"Smoother solution paths": We use "smoother" to describe paths where the norms of the weights change predictably as λ is varied. As shown in the bottom panels of Figure 1, this is in contrast to the large, erratic jumps in the estimated weights that are visible in the non-pathwise and forward-pathwise results (top panels).
>
> •	"Different x-axis": Using a different range for log(λ) is intentional and standard when comparing different penalty types. The absolute scale of λ is not directly comparable between penalties (e.g., the λ value that produces a 10-variable model for GMCP is different from the value for GLASSO). The purpose of these plots is to show the qualitative behavior of the solution path as it traverses from a dense to a sparse model.
>
> •	"Curves with the same color": We apologize for the confusion. Each individual line in the plot represents the solution path for a single input variable's group weight norm (∣∣W0,j∣∣2). The colors are used to group variables into two categories: informative variables (red) and nuisance variables (blue). There are four red lines because there are four true informative variables, and sixteen blue lines for the sixteen nuisance variables.
>
> Action Taken: We have substantially revised the caption for Figure 1 and the corresponding text in Section 3.2 to explicitly define "smoother paths," explain the x-axis, and clarify that each line represents a single variable, with colors indicating the variable type.
>
>
> 2. On the Stability of the Adam Method
>
> Reviewer's Comment: "According to Section 4, Adam method is used in numerical results as the optimization algorithm. Could you clarify in what sense it is stable and why?"
>
> Our Response: We thank the reviewer for this important clarifying question. As noted above, our claim of "stability" does not refer to a property of the Adam optimizer itself. Adam is used as the underlying optimizer to find a local minimum for each fixed value of the regularization parameter, λ. Instead, the stability we describe is a property of the solution path, which is generated by our backward path-wise optimization strategy. As we have now explicitly defined in the revised Section 3.2, a stable path is one where the model's parameters and sparsity level evolve predictably as λ changes. This is achieved by using the solution for one λ as a warm start for the next, which prevents the large, erratic jumps in the solution that would otherwise occur.
>
> Action Taken: We have revised Section 3.2 to clearly distinguish between the role of the Adam optimizer (finding a local minimum at each step) and the stability of the solution path that results from our backward path-wise strategy.

---

> ### Author Response · Authors · 2025-09-13
> **Authors' Response to Reviewer Comments (part 3)**
>
> 3. On the Comparison to SOTA
>
> Reviewer's Comment: "The comparison with SOTA in Section 5 seems to be of limited impact... At the current test accuracy level, linear SVM could also learn something like Fig 6. Why is this method not compared? What is the SOTA AUC performance in the survival dataset...?"
>
> Our Response: We thank the reviewer for raising this insightful point. The reviewer is correct that the performance of our MLP-based models on these specific tasks does not match the SOTA results achieved by highly specialized architectures. We have clarified in the manuscript that achieving SOTA performance was not the primary goal of these real-data examples. Instead, they serve as challenging case studies to demonstrate our method's properties in specific contexts.
>
> •	Regarding MNIST: We intentionally designed this experiment to be a high-dimensional, low-sample-size problem (p=784,n=500) to align with the focus of our paper. The goal was not to build the best image classifier, but to use MNIST as a visualizable high-dimensional feature selection task to see if the method could correctly identify the relevant pixels (features). We used a standard MLP to demonstrate the general-purpose capability of our feature selection mechanism, without relying on the strong architectural priors of a CNN.
>
> •	Regarding CALGB Survival Data: The goal here was to demonstrate the novel application of our sparse neural network framework to complex, high-dimensional time-to-event data, a domain where such methods are not commonly used, and to compare the behavior of different penalties within that framework.
>
> We agree with the reviewer's excellent suggestion that exploring more complex network architectures to close the performance gap to SOTA is an important direction for future work.
>
> Action Taken:
> (1) We have added a comprehensive introductory paragraph at the beginning of Section 6 that clarifies the overarching goal of the real-data examples and explains the specific purpose of the CALGB and MNIST case studies.
>
> (2) We have added a new paragraph to the Discussion (Section 7), placed just before the final concluding paragraph, which acknowledges that the exploration of more specialized architectures is a promising avenue for future research.

---

### Comment · Action_Editor_MeM1 · 2025-09-05
**Rebuttal phase extended**

Dear Reviewers: At the request of the authors, the rebuttal phase for this submission has been extended. The authors have until **September 12 2025** to submit their response to your reviews, after which discussions will commence.

Please allow the authors this extra time before making your official recommendations. If you have any questions, please let me know here.
-AE

---

### Decision · Action_Editor_MeM1 · 2025-10-20

**Recommendation:** Accept with minor revision

**Additional Comments:**

Two reviewers left comments requiring a minor revision. Please address the following issues in your revision:

1. The reference Lederer (2022) is not included in the references (it seems there may be a tex issue on p. 2)
2. Condition 4.3 assumes the exists of the third derivative. Does ReLU network satisfy this condition? Please add discussion on it.
3. The text still needs to be refined, e.g. Figure 1 caption mentions red curves which are in green.

**Audience:**

Yes

**Audience Explanation:**

In addition to the detailed theoretical and empirical study, which is a nice addition to the literature, the general problem of variable selection with neural networks should be of interest to some in the TMLR audience.

**Claims And Evidence:**

Yes

**Claims Explanation:**

All reviewers raised significant concerns during the initial review period, and the authors submitted a detailed and extensive revision that includes additional discussion and citations alongside an entirely new theory section. After revision and discussion, this submission clears the acceptance criteria for TMLR.